# Cytokine-armed pyroptosis induces antitumor immunity against diverse types of tumors

Sara Orehek [1,2], Taja Železnik Ramuta[1], Duško Lainšček [1,3,4], Špela Malenšek [1,2], Martin Šala [5], Mojca Benčina [1,4,6], Roman Jerala [1,3,4] & Iva Hafner-Bratkovič [1,3,7] ✉

Inflammasomes are defense complexes that utilize cytokines and immunogenic cell death (ICD) to stimulate the immune system against pathogens. Inspired by their dual action, we present cytokine-armed pyroptosis as a strategy for boosting immune response against diverse types of tumors. To induce pyroptosis, we utilize designed tightly regulated gasdermin D variants comprising different pore-forming capabilities and diverse modes of activation, representing a toolbox of ICD inducers. We demonstrate that the electrogenic transfer of ICD effector-encoding plasmids into mouse melanoma tumors when combined with intratumoral expression of cytokines IL-1β, IL-12, or IL-18, enhanced anti-tumor immune responses. Careful selection of immunostimulatory molecules is, however, imperative as a combination of IL-1β and IL-18 antagonized the protective effect of pyroptosis by IFNγ-mediated upregulation of several immunosuppressive pathways. Additionally, we show that the intratumoral introduction of armed pyroptosis provides protection against distant tumors and proves effective across various tumor types without inducing systemic inflammation. Deconstructed inflammasomes thus serve as a powerful, tunable, and tumor-agnostic strategy to enhance antitumor response, even against the most resilient types of tumors.

To protect the integrity of organisms and cells, the immune system evolved into one of the most complex networks. It is particularly effective in defense against pathogens, however, it often fails to mount equally potent responses against cancer, as tumors developed a multitude of ways to evade immunosurveillance and disrupt the cancer immunity cycle[1]. The main goal of cancer immunotherapy is to trigger immune responses directed against the tumor. These depend on the immunogenicity of the malignant cells, the ability to evoke adaptive immune responses, and the immunosuppressive tumor microenvironment (TME)[2]. Based on effector immune cell presence in the

TME, tumors can be divided into inflamed (hot), immune excluded, and immune desert (cold) tumors[3–5]. Cold tumors, which elude T cell priming and infiltration, are not amenable to immune checkpoint inhibitor (ICI) therapy[3], prompting the development of diverse strategies for stimulating immunity against this most resistant type of tumor.

The innate immune system with a lineup of genetically encoded pattern recognition receptors provides the fastest response to various types of threats and instructs the development of adaptive immune responses[6]. Inflammasomes, the cytosolic multiprotein platforms for activation of proinflammatory caspases[7], are powerful proinflammatory

[1]Department of Synthetic Biology and Immunology, National Institute of Chemistry, Ljubljana, Slovenia. [2]Interdisciplinary Doctoral Study of Biomedicine, Faculty of Medicine, University of Ljubljana, Ljubljana, Slovenia. [3]EN-FIST Centre of Excellence, Ljubljana, Slovenia. [4]Centre for the Technologies of Gene and Cell Therapy, National Institute of Chemistry, Ljubljana, Slovenia. [5]Department of Analytical Chemistry, National Institute of Chemistry, Ljubljana, Slovenia. [6]Biotechnical Faculty, University of Ljubljana, Ljubljana, Slovenia. [7]Faculty of Medicine, University of Ljubljana, Ljubljana, Slovenia. ✉e-mail: iva.hafner@ki.si

machines as their activation leads not only to the processing and release of proinflammatory cytokines IL-1β and IL-18 but also to pyroptosis, whose main executor is gasdermin D (GSDMD), a member of the gasdermin protein family[8,9]. Upon proteolytic cleavage by inflammasome-activated caspases in the linker region connecting the N- and C-terminal GSDMD domains, the released N-terminal domain (NT GSDMD) assembles into membrane pores[10–14], facilitating the release of intracellular components[15–17] and inducing pyroptosis, a type of immunogenic cell death (ICD). ICD represents a functionally unique response pattern that involves the release of damage-associated molecular patterns (DAMPs), tumor antigens, neo-antigens, and other immunostimulatory components from the cancer cells[18–20]. These diverse signals excite the infiltration of different immune cells into the TME and enable their efficient activation and maturation. An important mechanism of immune escape by cancer cells is the suppression of different cell death pathways either due to epigenetic downregulation or through loss-of-function mutations of the main cell death effectors such as gasdermins D and E[11,21–25]. Emerging evidence shows that exogenously induced gasdermin-mediated pyroptosis[23,26,27] can act as an antitumor remedy, not only by directly destroying cancer cells but also by modulating TME[28].

Inflammasome-dependent cytokines, IL-1β and IL-18, are proinflammatory cytokines with known functions in innate and adaptive immunity. IL-1β is an important orchestrator of adaptive immunity promoting CD4+ and CD8+ T cell responses and memory cell generation[29–33]. Furthermore, IL-18 is a major stimulator of IFNγ release from natural killer (NK) and type 1 helper T (Th1) cells[34]. It has a long track record as an antitumor cytokine and has been well tolerated in clinical trials, and even though those failed due to the lack of efficiency[35], recent preclinical studies revealed the therapeutic potential of this cytokine in combination with ICI[36] and CAR T cell adoptive therapy[37]. Several cytokines, unrelated to inflammasome signaling, such as IL-12[38–40], were shown to be potent inducers of antitumor responses making cytokine therapy an attractive approach for cancer immunotherapy.

Systemic immune responses are downregulated in the TME and even though necrotic cell death enhances the release of tumor antigens, it fails to stimulate multiple components of the immune network and efficient tumor inhibition[41,42]. Thus, especially for immunologically cold tumors that are poorly responsive to some already established treatments, new approaches that exploit innate and adaptive arms of the immune response are needed.

In this work, inspired by natural inflammasomes that drive potent defense responses mediated by cytokines as well as pyroptosis, we investigate whether local deconstruction of inflammasomes in tumors can induce efficient antitumor immunity. By combining both signaling arms of inflammasomes, the immunogenic cell death and the cytokine component, we aimed to create the so-called "cytokine-armed pyroptosis", with the intent to induce the production of immunostimulatory cytokines before the cells are lysed, releasing both cytokines as well as tumor antigens that trigger an adaptive immune response. Even when applied to relatively large tumors, we demonstrate the protective effect of electroporation of plasmids encoding pyroptotic component in the mouse melanoma model, which is further enhanced by local production of immunostimulatory component such as cytokine. Unexpectedly the protective effect of pyroptosis is lost when IFNγ or a combination of IL-1β and IL-18 is introduced. We further show that local deconstruction of inflammasomes induces potent anti-tumor immunity without causing systemic inflammation, protecting the mice from tumor rechallenge. Finally, our results demonstrate that deconstructed inflammasomes provide protection in diverse mouse tumor models.

## Results

### A toolbox of GSDMD variants for induction of pyroptosis
N-terminal domains of gasdermins are very potent inducers of cell death[8,9]. To explore the potential of gasdermin-facilitated pyroptosis

in cancer immunotherapy we chose gasdermin D as a prototypic inflammasome-connected gasdermin, whose pore-forming activity is boosted by oxidative stress[43,44], a characteristic of TME. Cancer cells are known to downregulate the expression of the components of cell death pathways[21–25]. We profiled three murine cancer cell lines: B16F10 (melanoma), 4T1 (breast cancer) and CT26 (colon cancer) for the expression of cell death effector gasdermin D (Supplementary Fig. 1a), and the components of NLRP3 inflammasome (Supplementary Fig. 1b). We demonstrate that each tested cell line is defective in the expression of at least one of the inflammasome components, suggesting those cells are not able to respond to inflammasome activators which prompted the development of synthetic pyroptotic systems.

For induction and regulation of pyroptosis in tumors, we devised GSDMD variants with different pore-forming efficacy and divergent modes of activation (Fig. 1a). GSDMD is dormant in the cytosol until proteases such as caspase-1 cleave it at position 276 releasing the N-terminal pore-forming domain[10–14]. In addition to NT GSDMD (1-276), we prepared a hypomorph variant NT GSDMD I105N carrying the isoleucine 105 to asparagine mutation that kinetically attenuates pore formation[14,15,43,44]. We also prepared GSDMD^TEV where the caspase-1/11 cleavage site was mutated into the Tobacco etch virus protease (TEVp) recognition site which enables controlled cleavage of GSDMD and induction of pore formation using TEVp. TEVp is a highly specific protease recognizing a sequence of seven amino acid residues, minimally interferes with the existing cellular chassis, and is nontoxic to mammalian cells[45], therefore suitable for use in the setup preferring the orthogonality facilitated by the non-endogenous enzyme.

To examine their pore-forming capability and the level of cell death induction, we transfected human embryonic kidney 293T (HEK293T) cells that do not express endogenous GSDMD with plasmids encoding GSDMD variants, bypassing upstream regulatory events that normally lead to pyroptosis. Cytolysis was measured by the release of the lactate dehydrogenase (LDH)[15–17], whereas pore formation was characterized by uptake of the membrane-impermeable dye propidium iodide (PI) (Fig. 1b). Although caspase-1 requires activation by inflammasomes, its overexpression leads to spontaneous activation[46] due to oligomerization through caspase activation and recruitment domain (CARD) filament formation[47]. Full-length GSDMD and GSDMD^TEV only induced pore-formation and cell death when the corresponding activating proteases were present (Fig. 1b). On the other hand, NT GSDMD and NT GSDMD I105N induced pore-formation and cell death spontaneously upon expression. All variants were effective at all concentrations tested and have reached the maximum efficiency already with the lowest concentration, except for NT GSDMD I105N where concentration dependence of pore-formation and LDH release could be observed (Fig. 1b).

Taken together, we prepared a panel of GSDMD-based cell death effectors with different modes of regulation and strength of cell death induction that can be applied in different situations where either instant or conditional cell death is preferred. Moreover, these systems act independently of the endogenous inflammasomes and are therefore suitable for use in versatile situations where inflammasome components are not expressed.

### Intratumoral delivery of NT GSDMD promotes tumor regression
We first explored the functionality of exogenously introduced GSDMD in a B16F10 cancer cell line that does not express either GSDMD or some other inflammasome components (Supplementary Fig. 1). Transfection of B16F10 cells with a construct encoding NT GSDMD induced the characteristic ballooning of the cells (Fig. 2a and Supplementary Fig. 2) and permeabilization of the plasma membrane (Fig. 2b). In contrast, no pyroptotic morphology and only a few PI-positive cells were observed in the untransfected and empty vector-transfected cells (Fig. 2a, b and Supplementary Fig. 2). We could not detect NT GSDMD in any other cell lysates but the ones transfected

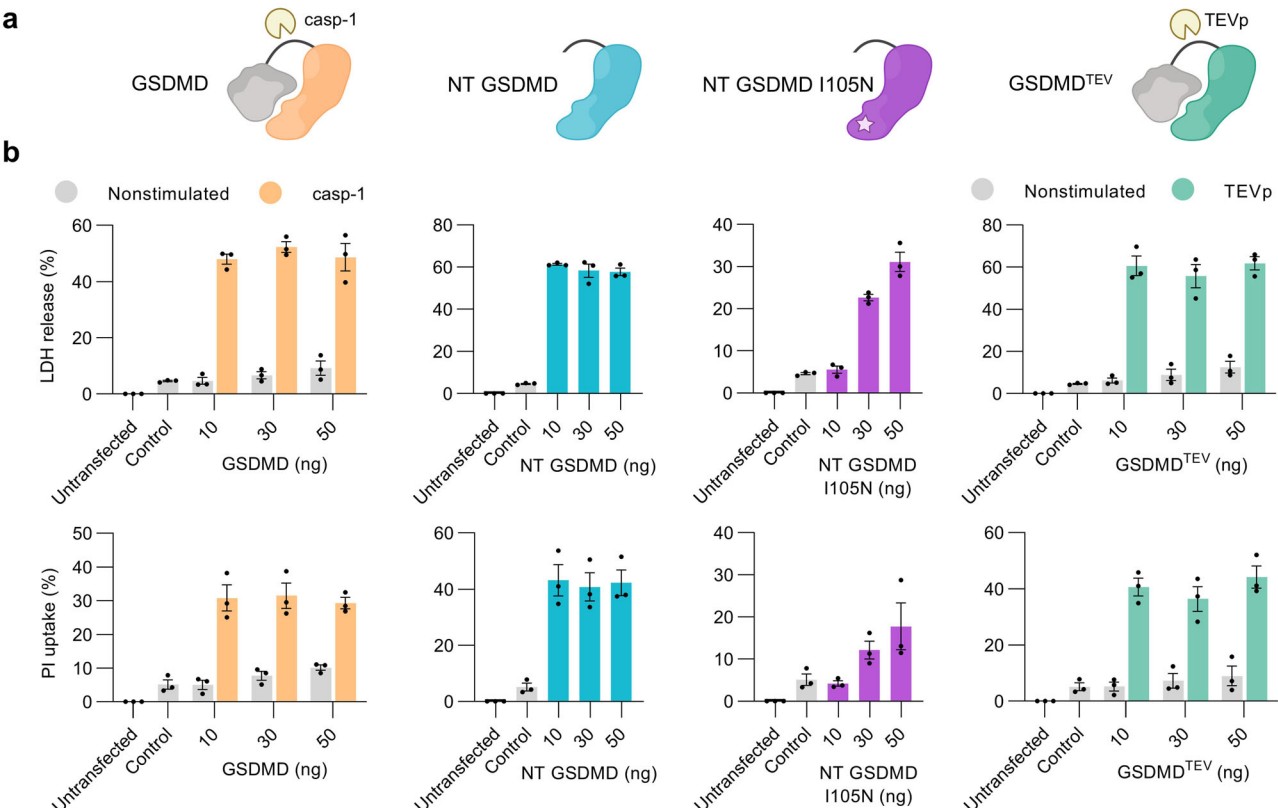

**Fig. 1 | Design and in vitro validation of GSDMD-induced cell death. a** Schematic of designed GSDMD variants with the C-terminal auto-inhibitory domain (gray) and N-terminal domain in different colors. The linker region harbors cleavage sites for various proteases. **b** In vitro cell death characterization. HEK293T cells were transfected with plasmids encoding different GSDMD variants or empty vector as a control and after 48 h lactate dehydrogenase (LDH) release and propidium iodide (PI) uptake were measured. Plots (**b**) show the means ± SEM of three independent experiments. Source data are provided as a Source Data file. Figure 1a created in BioRender. Hafner Bratkovic, I. (2024) https://BioRender.com/o17o904.

with NT GSDMD (Fig. 2c) thus correlating the expression of the toxic protein with pyroptosis.

Next, we evaluated the impact of GSDMD-induced pyroptosis on tumor growth in the B16F10 model in vivo, a known model of immunologically cold tumors[48]. In our experimental system, we subcutaneously inject B16F10 cells and afterward treat middle- to large-sized tumors (5–9 mm longest dimension) via electrogenic transfer of plasmid DNA (Supplementary Fig. 3a). We assessed the effectiveness of electrogenic transfer into the established B16F10 melanoma tumors using the yellow fluorescent protein (YFP)-coding plasmid. Approximately 3.5% of YFP-positive cells were detected (Supplementary Fig. 3b), and the large majority (95%) of which were melanoma cells (Supplementary Fig. 3c). For the assessment of pyroptosis-induced tumor surveillance we employed NT GSDMD, therefore, omitting the need for an additional cleavage of full-length GSDMD. Intratumoral injection (i.t.) followed by electroporation of a plasmid encoding NT GSDMD (Fig. 2d) resulted in tumor regression and prolonged survival (Fig. 2e, f and Supplementary Fig. 3d) compared to the control group where tumors were electroporated with empty vector. The survival of the control group was similar to the untreated group, indicating that plasmid electroporation by itself and the potential activation of DNA innate immune sensors by plasmid DNA do not impact tumor growth and animal survival (Fig. 2e, f). Additionally, we tested the outcome of i.t. injection of plasmids unaccompanied by electroporation, addressing the capacity of plasmid gymnosis by cancer cells. All tumors were unaffected by the treatment, suggesting that plasmid gymnosis is insufficient for encoded construct expression, therefore plasmid electroporation is a key feature of the proposed treatment (Supplementary Fig. 3e).

NT GSDMD therapy not only increased the survival time but also led to full remission in approximately 20% of treated animals and an extension of tumor-free survival with no recurrence to 500 days when mice were eliminated due to old age. This suggests that therapeutically applied GSDMD acts as a tumor suppressor. The pyroptotic capacity of in-situ electroporated NT GSDMD was confirmed with an increased number of PI-positive tumor cells in the NT GSDMD-treated tumors compared to empty vector-treated control (Fig. 2g and Supplementary Fig. 4a, b).

Our results demonstrate that inducing GSDMD-mediated pyroptosis in immunologically silent tumors leads to tumor regression and improves long-term, tumor-free survival in treated mice.

## On-demand small molecule-controlled tumor pyroptosis facilitated by engineered GSDMD

We reasoned that additional control over the timing of tumor cell pyroptosis induction might be needed in cases of combinatorial therapy where immunostimulatory genes encoding e.g., cytokines would be introduced, or to impede adverse side effects such as cytokine release syndrome. Thus, we developed two systems that trigger GSDMD-based pyroptosis upon the addition of a small molecule utilizing ligand-dependent dimerization (Fig. 3). The first system is based on chemically regulated split TEVp, with complementary split fragments fused to FKBP and FRB dimerization domains, whose heterodimerization is inducible using rapamycin or its analog (rapalog) AP21967 (Fig. 3a)[45]. The pore-forming capacity was assessed using LDH and PI assays upon transfection of constructs into HEK293T cells (Fig. 3b). While small molecule heterodimerizers were not toxic to cells and transfection of constructs encoding GSDMD[TEV]- split TEVp system

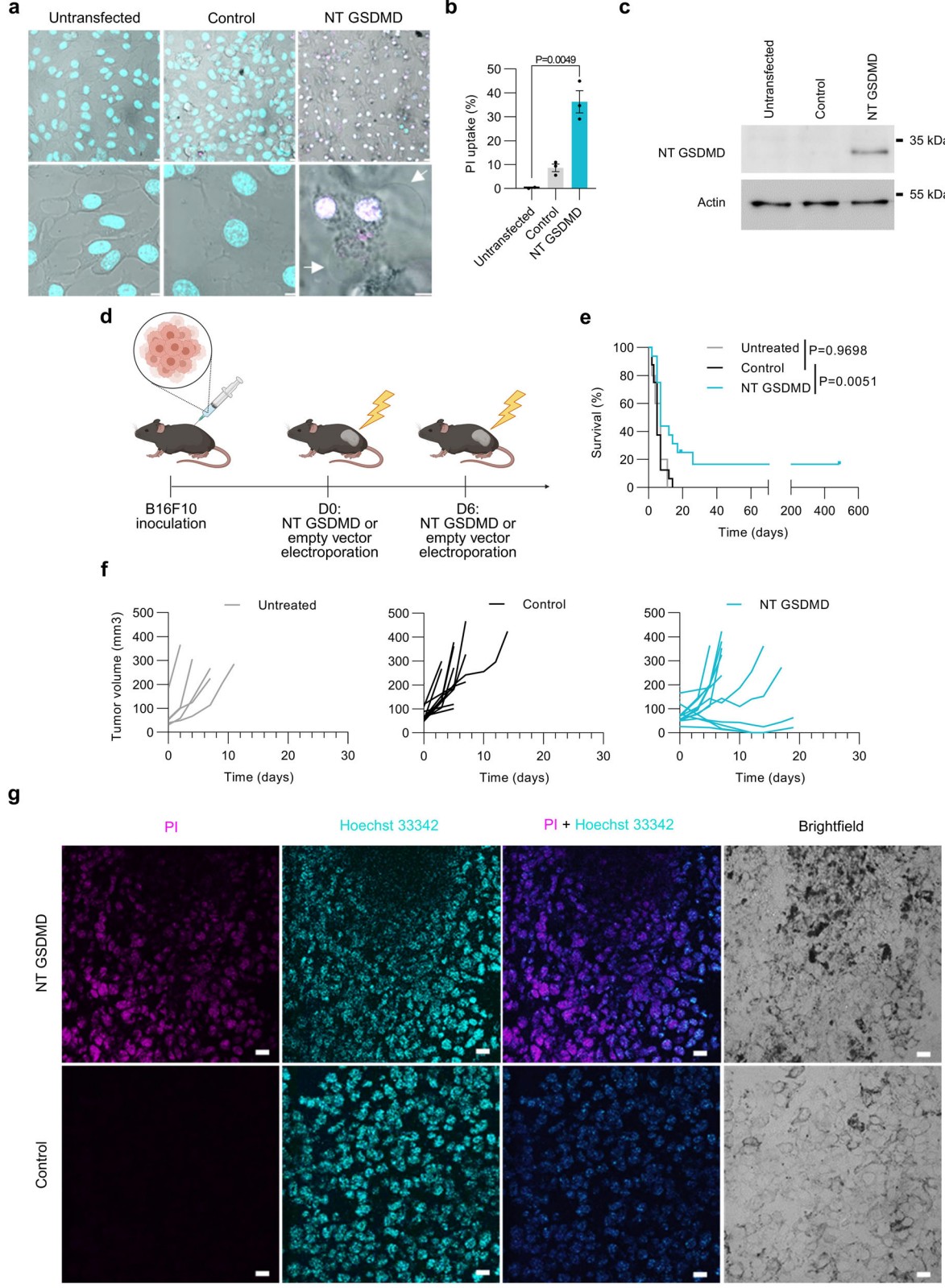

had a negligible effect in the absence of these ligands, the addition of rapamycin and AP21967 induced robust pore formation and LDH release (Fig. 3b), demonstrating the functionality of the developed system for induction of cell death using small molecule compounds, some of which (e.g., rapamycin) are approved for human clinical use.

While the GSDMD^TEV-split TEVp system is very specific and does not cleave endogenous GSDMD, it requires the simultaneous introduction of three constructs hampering its translational potential. We thus developed another system utilizing homodimerizing DmrB domains to induce self-association and activation of caspase-1 molecules missing the CARD. This construct should initiate cleavage of either endogenous or introduced GSDMD, mediated by chemical input signal - small molecule AP20187 (Fig. 3c)[49]. From the designs including single and double DmrB domains fused to casp-1ΔCARD[43], the one

**Fig. 2 | NT GSDMD treatment results in tumor surveillance.** In vitro pyroptosis characterization in B16F10 cells. **a** Confocal images of B16F10 cells transfected with NT GSDMD or empty vector as a control and stained with Hoechst 33342 (cyan) and PI (magenta). White arrows indicate ballooning cells. Scale bars represent 5 μm. **b** B16F10 cells were transfected with plasmid encoding NT GSDMD or empty vector and after 24 h PI uptake was measured. **c** Western blot analysis of the NT GSDMD expression in B16F10 cell lysates prepared 24 h after the transfection. **d–g** NT GSDMD-induced pyroptosis in vivo. **d** Experimental scheme. B16F10 tumor cells were s.c. implanted in C57BL/6J mice and i.t. injected with 20 μg plasmid encoding NT GSDMD (*n* = 16) or 20 μg empty vector as a control (*n* = 16), followed by electroporation. Tumors of untreated mice (*n* = 5) were not injected or electroporated.

**e** Kaplan-Meier plot represents the effectiveness of NT GSDMD in preventing tumor growth. **f** Tumor volume progression depicted for the individual mouse. **g** In vivo PI assay. Tumor-bearing mice were electroporated with NT GSDMD or empty vector 24 h prior to i.t. injection of PI. Representative tumor sections are shown. Scale bars 20 μm. Representative of three independent experiments is shown (**a, c**). Data (**b**) are presented as mean ± SEM of three independent experiments analyzed by a two-tailed unpaired Student's *t*-test. Kaplan-Meier curves (**e**) combine data from two independent experiments and were analyzed using log-rank test. Time (**e, f**) is defined as days post-first treatment. Source data are provided as a Source Data file. Figure 2d created in BioRender. Hafner Bratkovic, I. (2024) https://BioRender.com/z80o211.

utilizing double DmrB (dDmrB) outperformed the other as no basal activity in the absence of the homodimerizer was observed (Supplementary Fig. 5a). These results show that we established a system where the ligand induces homodimerization and activation of caspase-1. We further engineered a minimal tumor-agnostic inducible system for chemically controlled triggering of GSDMD-mediated pyroptosis, that would enable system introduction with a single electroporation, which would be beneficial for therapeutic use. We validated the designs with different orientations of the polycistronic construct including dDmrB casp-1- and GSDMD-encoding sequences, where dDmrB casp-1:GSDMD turned out to be more robust with negligible pore formation and cell death induction in the absence of AP20187 (Fig. 3c, d), whereas GSDMD:dDmrB casp-1 showed extensive leakage with the increasing amount of plasmids transfected and almost the same percentage of cell death even without the addition of AP20187 (Supplementary Fig. 5b, c). Therefore, dDmrB casp-1:GSDMD was first tested in vitro in B16F10 cells, where the addition of AP20187 triggered GSDMD cleavage (Supplementary Fig. 5d) and consecutive membrane permeabilization (Supplementary Fig. 5e). Finally, this construct was further evaluated in the B16F10 melanoma mouse model (Fig. 3e–g). The treatment where plasmid electroporation was followed by AP20187 injection induced prolonged survival (Fig. 3f, g) and long-lasting remission (Supplementary Fig. 5f). In the absence of AP20187 or with AP20187 treatment without our inducible construct, no antitumor efficacy was discerned (Fig. 3f, g). None of the groups showed a reduction in body weight (Supplementary Fig. 5g) or other changes in the biochemical analysis of the blood samples (Supplementary Fig. 5h).

We developed a minimalistic small molecule-induced system that enables potent, tumor expression profile-independent induction of GSDMD-mediated pyroptosis and antitumor responses.

## Deconstruction of the inflammasome responses in cancer cells enhances tumor clearance

Inflammasome assembly is followed by multiple significant events in addition to GSDMD pore formation. Simultaneously to GSDMD cleavage, activated caspase-1 cleaves the precursors of cytokines IL-1β and IL-18 into their active form. These cytokines act as the amplifier of the immune reactions and have an established function in the modulation of the innate immune response and T-cell mediated antitumor immunity[30,35]. Therefore, we hypothesized that an artificial deconstruction of the inflammasome responses in the tumor cells, consisting of a combination of proinflammatory cytokines and cell lysis, could further potentiate antitumor immunity.

Before testing the idea in vivo, HEK293T cells were cotransfected with plasmids encoding mature forms of IL-1β or IL-18 and GSDMD variants to appraise the capacity of transfected cells to produce cytokines before dying (Fig. 4a). Cotransfection of the cytokines did not affect caspase-1-cleaved GSDMD nor NT GSDMD-facilitated pore formation and cytolysis. Mature IL-1β can translocate through the membrane by binding to phosphatidylinositol 4,5-bisphosphate (PIP2)[50], thus transfection of plasmid encoding mature IL-1β led to the detection of secreted IL-1β even in the absence of GSDMD pores and was enhanced when GSDMD was cleaved by caspase-1 and formed

pores (Fig. 4a). IL-18 on the other hand was only released in the cell supernatant upon initiation of GSDMD pore formation. In the case of NT GSDMD, we observed a perceptible decrease in the concentration of both cytokines in the supernatant compared to caspase-1-cleaved GSDMD, suggesting that rapid cell death induced by NT GSDMD inhibits cytokine expression (Fig. 4a).

Guided by these results, we generated an in vivo experimental outline where plasmids encoding cytokines were electroporated into the tumors two days before the one encoding NT GSDMD, which allowed for cytokine expression prior to pyroptosis of cytokine-producing cells (Fig. 4b). Although previous reports using recombinant protein showed that IL-1β can act as a tumor-suppressing agent in B16F10 and other tumor types[51,52], single electroporation of plasmid encoding IL-1β did not exhibit any tumor shrinkage capacity nor long-term survival (Fig. 4c, d and Supplementary Fig. 6a), likely due to lower IL-1β exposure or other parameters originating from different treatment systems used. However, we further demonstrated that deconstruction of inflammasome with cytokine followed by GSDMD-mediated pyroptosis potentiated antitumor treatment led to 40% long-term tumor-free survival observed for IL-1β and 37.5% for IL-18 (Fig. 4c, d and Supplementary Fig. 6a) compared to electroporation of an empty vector that did not affect tumor regression (Fig. 4c, d).

To build up a complete deconstructed inflammasome, we combined pyroptosis with both inflammasome-dependent cytokines (NT GSDMD + IL-1β + IL-18). Expecting an additive or synergistic effect on boosting antitumor immunity, we were surprised that instead of enhancing antitumor responses, the combination of cytokines IL-1β and IL-18 even abolished the effect of pyroptosis and the response did not differ from the control (Fig. 4c, d).

In line with these results, we conclude that pyroptosis of cancer cells is crucial for eliciting potent antitumor responses that lead to tumor eradication and long-term animal survival and this effect can be enhanced by an ectopic expression of proinflammatory cytokine.

## IL-12 enhances the tumor clearance capacity of pyroptosis

Encouraged by the potentiation of antitumor responses driven by GSDMD-induced pyroptosis in combination with inflammasome-related cytokines, we wanted to further explore the use of other cytokines in combination with cell lysis. We designed a combinatorial immunotherapy approach exploiting adjuvant attributes of cytokine IL-12, especially its tumor-suppressive characteristics and the ability to prime TME[53], hence creating an immune-supportive milieu. For that, we employed a single-chain IL-12 fusion protein of p40 and p35 subunits linked together[54]. In vitro screening of designed IL-12 fusion protein in combination with GSDMD variants revealed, as already shown with IL-1β and IL-18 (Fig. 4a), that the cotransfection of plasmids encoding the cytokine with cell death-inducing constructs does not impact cytotoxicity (Fig. 5a), however, cotransfection of NT GSDMD-encoding plasmid decreased secretion of IL-12 due to rapid cell death (Fig. 5a).

We further evaluated whether single intratumoral electrogenic treatment of IL-12-coding plasmid would induce antitumor responses and improve tumor control accomplished by NT GSDMD evoked

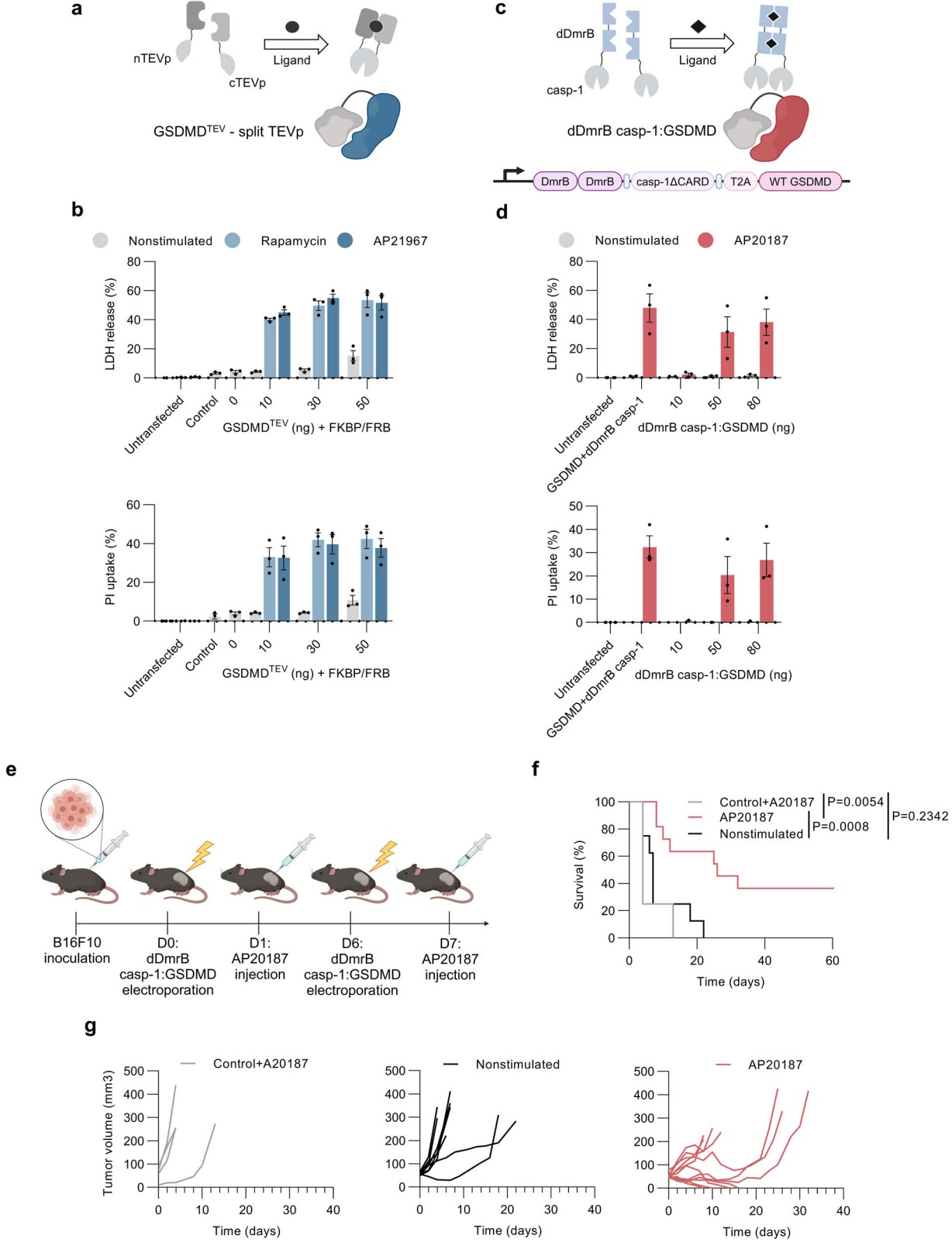

pyroptosis. The same experimental timeline was used as for the previous in vivo experiments (Fig. 5b). Electrotransfer of a single dose of IL-12 with consecutive electroporations of an empty vector did not lead to long-lasting remission, instead, all mice were eliminated by day 23 (Supplementary Fig. 6b). Together with the data obtained from the administration of IL-1β alone (Fig. 4c) and in line with previous studies employing several electroporations[55,56], we find that single

electroporation of cytokine-encoding plasmid is not sufficient to achieve antitumor effect that could lead to a progressive tumor volume decrease and eventual eradication, suggesting that in this experimental setting, induction of pyroptosis is crucial for achieving tumor growth control. However, IL-12 significantly enhanced the antitumor effects of NT GSDMD-induced pyroptosis and extensive control of tumor growth (Fig. 5c, d and Supplementary Fig. 6c). In

**Fig. 3 | Engineered GSDMD enabling inducible activation of GSDMD-prompted pyroptosis. a, c** Schematic of engineered GSDMD variants. **a** GSDMD^TEV with caspase-1/11 cleavage site mutated into the TEVp recognition site is activated upon ligand-mediated reconstruction of split TEVp protease (nTEVp and cTEVp). **c** Bicistronic construct consisting of WT GSDMD coupled with caspase-1 ΔCARD fused to double ligand-mediated homodimerization DmrB domains. **b, d** Characterization of the inducible systems in vitro. Ligand-dependent dimerization of split TEVp and dDmrB casp-1 and consecutive GSDMD activation was investigated in HEK293T cells by measuring LDH release and PI uptake 24 h after induction of dimerization with small molecules rapamycin (100 nM), AP21967(100 nM) or AP20187 (500 nM). **e** Experimental scheme. Mice from nonstimulated (n = 8) and AP20187 stimulated (n = 11) groups were electroporated with dDmrB casp-1:T2A:GSDMD plasmid (20 µg) and next-day dimerization of the DmrB domains was induced by i.t. administration of AP20187 in PBS, whereas PBS was administrated to

the nonstimulated group. The control group (n = 4) was electroporated with an empty vector and injected with AP20187 24 h after. **f** Kaplan-Meier plot showing the difference in survival of combined data from two independent experiments, analyzed by log-rank test. **g** Tumor growth curves of individual mice. Data (**b, d**) is presented as mean ± SEM of three independent experiments. Time (**f, g**) is defined as days after the first treatment. Source data are provided as a Source Data file. Figure 3a created in BioRender. Hafner Bratkovic, I. (2024) https://BioRender.com/y23k939 and BioRender. Hafner Bratkovic, I. (2024) https://BioRender.com/o17o904; Fig. 3c created in BioRender. Hafner Bratkovic, I. (2024) https://BioRender.com/o17o904, BioRender. Hafner Bratkovic, I. (2024) https://BioRender.com/y23k939, and BioRender. Hafner Bratkovic, I. (2024) https://BioRender.com/t10w614; Fig. 3e created in BioRender. Hafner Bratkovic, I. (2024) https://BioRender.com/e42l963.

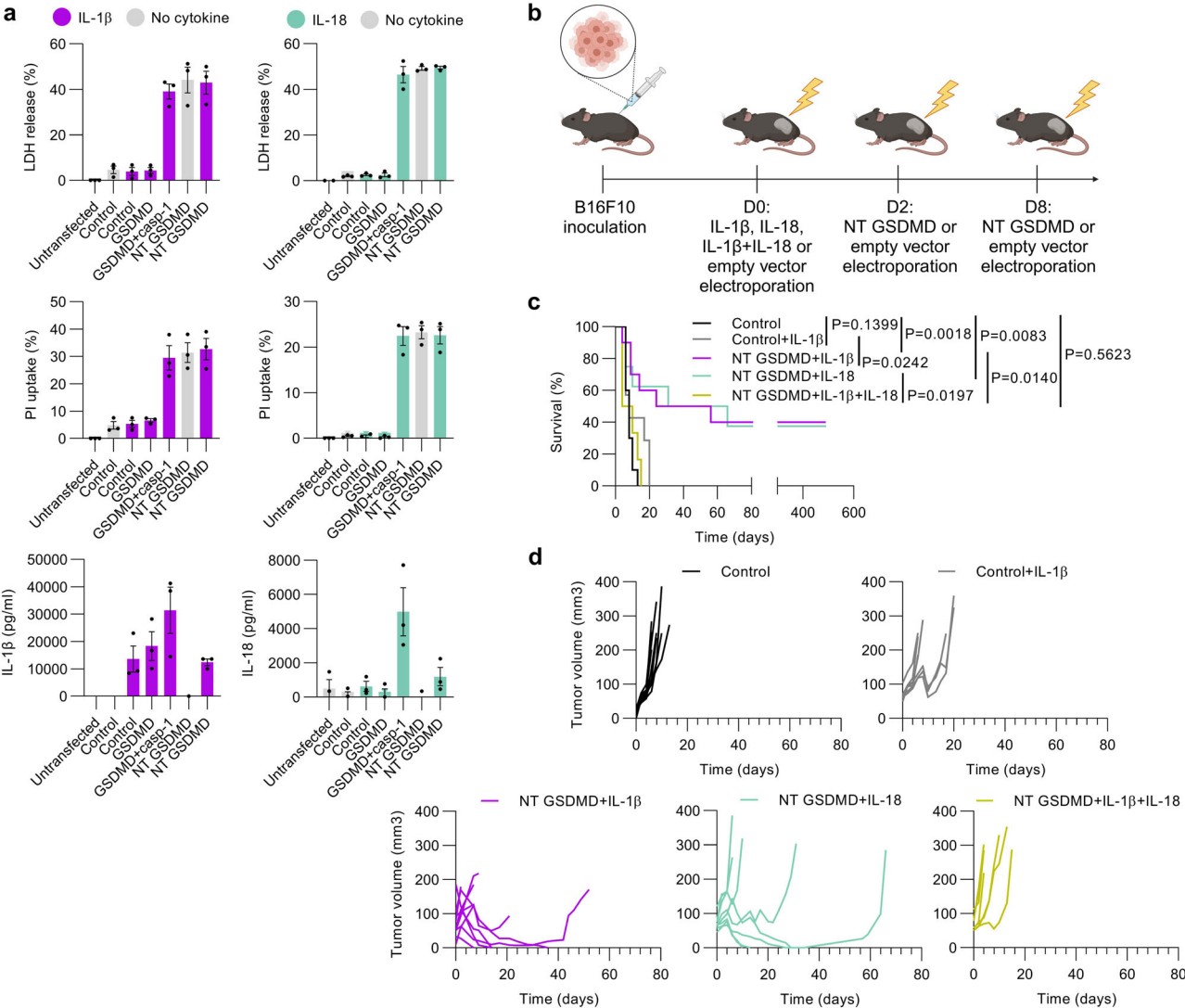

**Fig. 4 | Deconstruction of the inflammasome in cancer cells induces potent antitumor effects. a** Impact of the synthetic inflammasome on cell death and cytokine production in vitro. IL-1β and IL-18 cytokines were transfected together with GSDMD variants whose pore-forming activity was assessed with LDH release and PI uptake and cytokine concentration were determined by ELISA. **b** Experimental scheme. B16F10 cells were injected and tumors were electroporated with 20 µg IL-1β (n = 10), IL-18 (n = 8) or a combination of plasmids coding for IL-1β and IL-18 (10 µg each) (n = 6) 2 days before the first NT GSDMD

electroporation (20 µg). Empty vector electroporation was used as a control (n = 10) and in combination with IL-1β (n = 7). Survival (**c**) and tumor volume measurement (**d**) show the difference in tumor surveillance between various treatments. Data (**a**) are presented as mean ± SEM of three independent experiments. Time (**c, d**) is defined as days after the first treatment, and Kaplan-Meier curves (**c**) were analyzed with log-rank test. Source data are provided as a Source Data file. Figure 4b created in BioRender. Hafner Bratkovic, I. (2024) https://BioRender.com/x35z981.

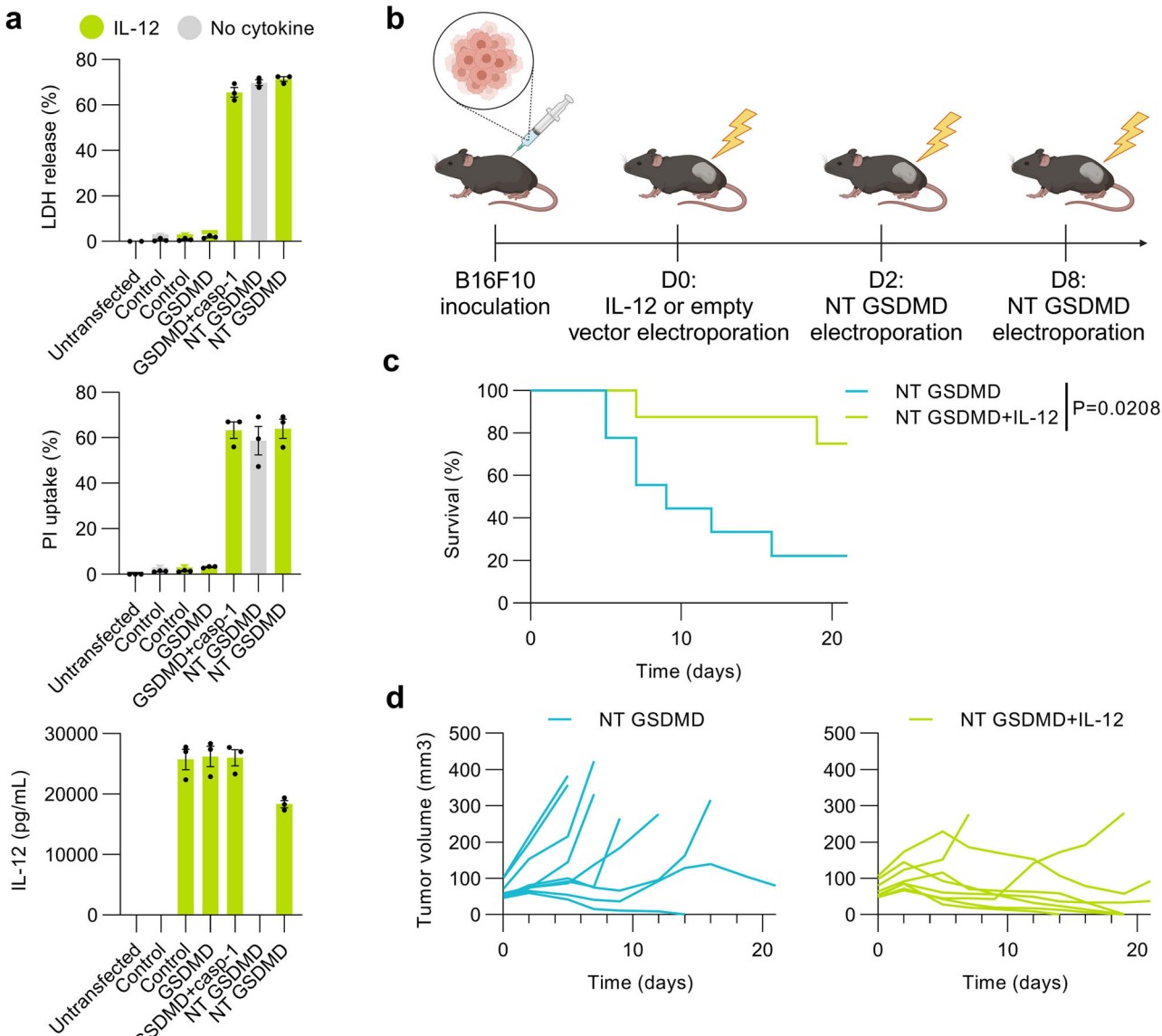

**Fig. 5 | The effect of IL-12 cytokine-armed pyroptosis in B16F10 mouse melanoma model. a** In vitro screening of single-chain IL-12 fusion protein in combination with GSDMD variants. Plasmids encoding IL-12 and GSDMD variants were transfected into HEK293T cells and pyroptotic capability of GSDMD variants was evaluated utilizing LDH release and PI uptake. IL-12 concentration was determined by ELISA. **b** Experimental scheme. Tumor-bearing mice were electroporated with 20 μg plasmid coding for single-chain IL-12 fusion protein (*n* = 8) or an empty vector (20 μg) and 2 days later first NT GSDMD plasmid (20 μg) electroporation followed. Data (**a**) are presented as mean ± SEM of three independent experiments. Survival chart (**c**) analyzed using log-rank test and tumor growth (**d**) following IL-12 cytokine-enriched pyroptosis. Time (**c, d**) is defined as days succeeding the first treatment. Source data are provided as a Source Data file. Figure 5b created in BioRender. Hafner Bratkovic, I. (2024) https://BioRender.com/e80o875.

comparison to the empty vector-treated mice that were rapidly eliminated, tumors were eradicated and long-lasting remission was achieved in more than 40% of mice treated with IL-12 and NT GSDMD (Supplementary Fig. 6d).

Systemic administration of IL-12 was reported to lead to significant adverse side effects limiting its suitability for therapy[53]. We were interested in whether the electrogenic transfer of DNA-encoding cytokines leads to a systemic increase in their concentration. The sera were tested for the presence of IL-1β, IL-18, or IL-12 cytokines 8 h after electroporation. Only IL-18 could be detected, but no increase in the IL-18 concentration was observed between the IL-18-treated group and other groups (Supplementary Fig. 6e). We also analyzed sera collected before the first treatment and 24 h after the first NT GSDMD electrotransfer for the presence of IFNγ and IL-6 (Supplementary Fig. 6f). The majority of sera samples had no detectable IFNγ and very low

concentrations of IL-6. These results suggest that i.t. electrogenic transfer of cytokine-encoding plasmids does not increase systemic cytokine levels. Additionally, cell death induced by NT GSDMD did not lead to systemic inflammation as judged by low and unchanged IL-6 levels.

The combinatory therapy of immunostimulatory cytokines IL-1β, IL-18, or IL-12 with GSDMD-based pyroptosis while not causing systemic adverse effects generates a robust antitumor response, ultimately leading to long-term remission in mouse melanoma model.

**Tumor histology reveals notable differences between effective and nonfunctional treatments**

Why does the single cytokine-armed pyroptosis eradicate tumors while NT GSDMD + IL-1β + IL-18 treatment does not exhibit any protection? To elaborate on this perplexing result, we first showed that

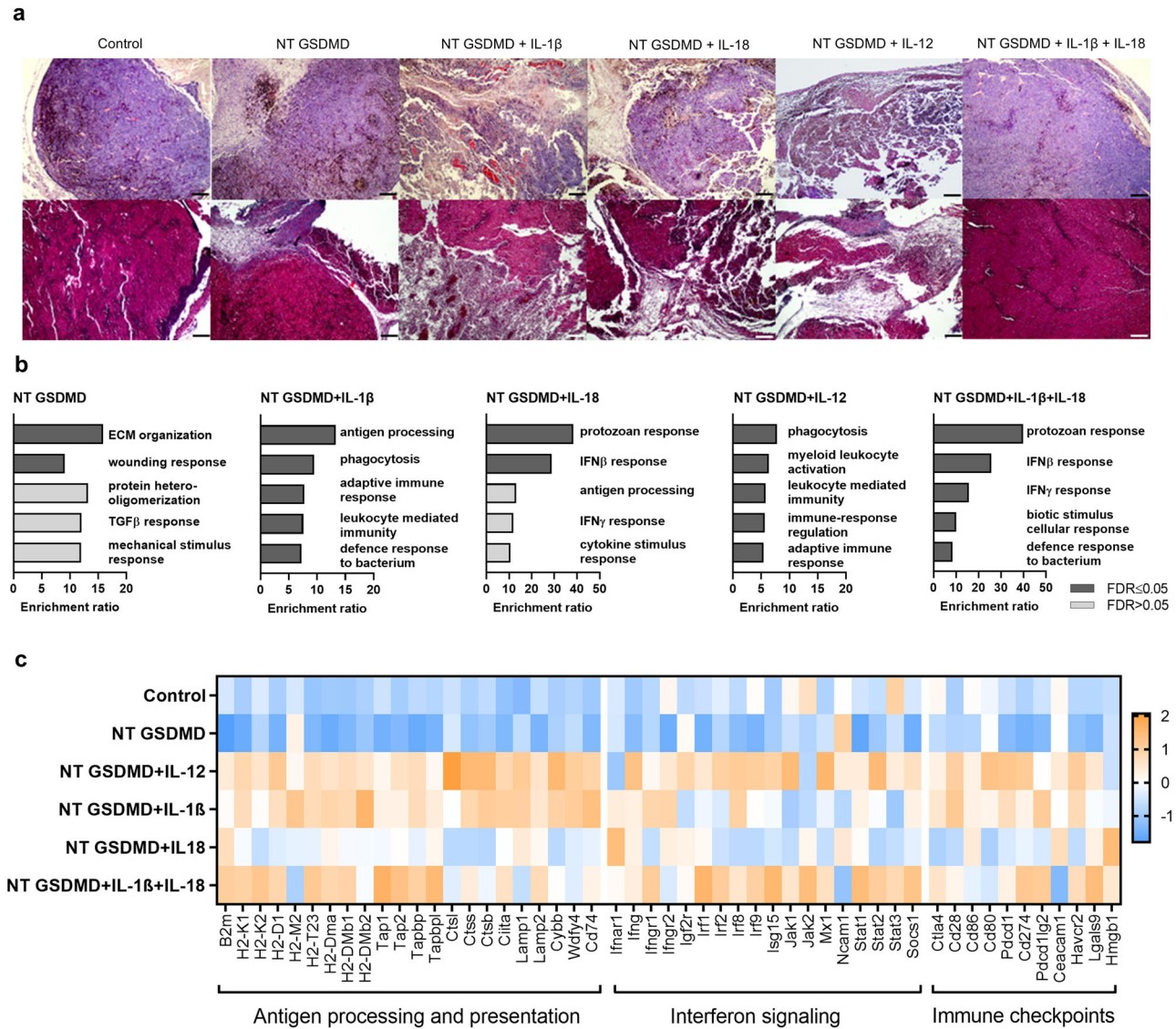

**Fig. 6 | Whole-tumor transcriptome analysis of different cytokine-armed pyroptosis treatments. a** Representative hematoxylin-eosin stain (top) and trichrome stain (bottom) of tumor samples treated with different ICD-inducing therapies. Images are representative of at least three different tumor samples per group from two different experiments. Scale bars 200 μm. **b** Enrichment results of over-representation analysis (ORA) from bulk B16F10 melanoma tumor RNA-seq analysis ($n = 3$ per sample), showing enrichment ratio of top 5 gene ontology terms for significantly upregulated genes of each sample (log2FC > 0.5, $p$adj <0.1, determined by wald test with DESeq2). Fisher's exact test was used to analyze the uploaded gene list for each sample. The mouse genome was used as the reference set. Benjamini-Hochberg test was used for multiple test adjustments, enriched categories were identified using the TOP method. **c** Differential gene expression. The heatmap shows selected genes involved in antigen processing and presentation, interferon signaling, and immune checkpoints. Z-scores are calculated over all samples, using DESeq2-normalized data. Source data are provided as a Source Data file.

electroporation of plasmids encoding IL-1β, IL-18, or IL-1β + IL-18 does not affect the B16F10 cell line proliferation in comparison to the control (Supplementary Fig. 7a). Next, we demonstrated that primary cultures from treated tumors did not significantly differ from the controls in terms of cell proliferation (Supplementary Fig. 7b), migration (Supplementary Fig. 7c), and morphology (Supplementary Fig. 7d). These results indicate that treatments do not alter the proliferative properties of cancer cells. Further, we investigated whether the reason for the absence of antitumor responses lies in the dosage of the electroporated cytokine DNA. In each electrogenic transfer, we introduce 20 μg of DNA, meaning that in the case of a combination of cytokines (IL-1β + IL-18), the amount of plasmid encoding each cytokine was halved (10 μg for each cytokine). The survival percentage of mice treated with a half-dose of IL-1β together with NT GSDMD was identical to the response obtained with the full amount

(Supplementary Fig. 7e, f and Fig. 4c), demonstrating that this is not the reason for the poor antitumor performance of the treatment with both inflammasome-dependent cytokines. Importantly, this experiment also suggests that upon optimization, lower amounts of therapeutic DNA could be used without hampering antitumor responses of armed pyroptosis treatment. To gain more insight into tissue histology, we conducted hematoxylin-eosin (HE) and trichrome staining. Stained tumor sections revealed large heterogeneity between different treatment groups (Fig. 6a and Supplementary Fig. 7g). Treatment strategies that were the most effective in tumor destruction, namely NT GSDMD combined with IL-1β, IL-18, or IL-12, showed similar tissue characteristics. These treatments prompted strong tumor reduction and degeneration with tissue destruction and an increased proportion of loose connective tissue and vascularization. In tumor sections, the tumor capsule was only partially visible, and in some parts of the tumor

hyperemia, coagulative necrosis, or hydropic degeneration could be perceived. On the other hand, samples from the control group and NT GSDMD + IL-1β + IL-18 displayed great resemblance in tissue structure, with mainly dense connective tissue that was in some parts interrupted with more loose connective tissue, demonstrating that this treatment only poorly affected tumor tissue architecture. Based on histological results, it is clear, that therapies combining pyroptosis with immunomodulatory properties of cytokines elicit changes in tumor tissue organization and eventually cause tumor deterioration.

### Tumors undergoing NT GSDMD + IL-1β + IL-18 treatment exhibit several immunosuppressive traits

We were interested in whether transcriptome analysis could provide insight into the signaling pathways activated in the tumor after each cytokine-armed pyroptosis therapy. We collected tumor samples at the time-point when control-treated tumors reached the defined humane endpoints, and analyzed tumor transcriptome with RNA sequencing. We were thus not interested in the immediate effects of cytokines but aimed to gather information on pathways that were stimulated in effective and ineffective treatments. Each treatment provided a unique pattern of significantly upregulated and downregulated genes compared to the control (Supplementary Fig. 8a, b), with a substantial proportion of these genes involved in biological processes of innate and adaptive immunity (Fig. 6b, c), such as antigen processing, cytokine stimulus-response, leukocyte mediated immunity, and IFN response. Downregulation of MHC I molecules and antigen presentation was shown to be one of tumor surveillance escape mechanisms[57–60]. However, this is not the reason for the observed lack of antitumor protection of deconstructed inflammasome employing both inflammasome-dependent cytokines, as we perceived a strong upregulation of transcripts for genes involved in MHC I molecule presentation (Fig. 6c and Supplementary Fig. 9a). The loss of MHC I presentation was previously attributed to defects in interferon pathway[59,61–64], as many molecules involved in MHC presentation are upregulated by interferon signaling. In agreement with the upregulation of MHC, we also observed a strong enrichment in interferon signaling for effective tumor-suppressing treatments, particularly for NT GSDMD + IL-12, as well as for ineffective NT GSDMD + IL-1β + IL-18 treatment (Fig. 6c and Supplementary Fig. 9b), although not necessarily with the same genes. Additionally, genes, associated with inflammasome and cell death including tumor suppressive TNF signaling pathways[65,66] were upregulated in the samples treated with NT GSDMD + IL-1β + IL-18 (Supplementary Fig. 10a). Even though those pathways when downregulated enable tumor escape and resistance to ICI therapy, a recent study demonstrated that interferon signaling can also enable immune suppression[67]. Dubrot et al. showed that IFNγ signaling and surface expression of classical MHC I molecules such as H2-K and H2-D (which were increased in NT GSDMD + IL-1β + IL-18-treated tumors) inhibits NK cell cytotoxicity[67,68]. Furthermore, we observed upregulation in *H2-T23* coding for Qa-1[b], a non-classical MHC class I molecule, which is IFNγ upregulated and inhibits cytotoxic CD8+ T cells[67].

Tumor cells can also suppress lymphocyte-mediated killing by upregulation of the immune checkpoints. We observed upregulation of transcripts encoding several immune checkpoints in deconstructed inflammasome-treated samples (Fig. 6c and Supplementary Fig. 10b). There was overexpression of *Havcr2* gene coding for TIM3 in both NT GSDMD + IL-12 and NT GSDMD + IL-1β + IL-18 samples, whereas its binding partner *Lgals9* (Galectin 9) was only upregulated in NT GSDMD + IL-1β + IL-18 samples. Furthermore, genes coding for immunostimulatory receptors expressed on T cells, e.g., *Ox40* and *Cd137*, were upregulated in NT GSDMD and NT GSDMD + IL-12-treated samples (Supplementary Fig. 10b). We also observed the upregulation of *Irf1*, a pivotal transcription factor for activation of PD-L1 expression and its ectopic expression is sufficient for PD-L1 upregulation even in

the absence of IFNγ[69,70]. Immunosuppressive genes *Cd274*, encoding PD-L1, and *Pdcd1lg2*, encoding PD-L2 checkpoints, can be upregulated by IFNγ signaling[71] and were in pair uniquely upregulated in tumors treated with abortive NT GSDMD + IL-1β + IL-18 treatment (Fig. 6c).

Cytokine-armed pyroptosis induces distinct late tumor transcriptome phenotypes with enrichment in multiple immune signaling pathways that inhibit tumor growth, however, IFNγ signaling can also lead to enhanced transcription of several immunosuppressive molecules.

### NT GSDMD + IL-1β + IL-18 treatment antagonizes tumor suppression through interferon signaling and immune checkpoint upregulation

This strong interferon fingerprint in NT GSDMD + IL-1β + IL-18-treated tumors prompted us to test whether IFNγ antagonizes the effect of NT GSDMD-mediated pyroptosis. While in vitro HEK293T cells effectively secreted IFNγ upon transfection, the overexpression of IFNγ did not impact the cytotoxicity of co-expressed NT GSDMD (Fig. 7a). Although IFNγ mounts great antitumor response when its production is triggered indirectly (e.g., via IL-12 or IL-18), the overexpression of IFNγ by cancer cells before cell death induction resulted in an abolished antitumor effect (Fig. 7b–d and Supplementary Fig. 11a) corroborating with the RNA sequencing results and in agreement with the reports suggesting IFNγ signaling-mediated suppression of antitumor responses[67,68].

The upregulation of several immune checkpoints, particularly both PD1 ligands in NT GSDMD + IL-1β + IL-18-treated tumors suggested that therapy with anti-PD1 antibody might be able to overcome the observed suppression of antitumor immunity. As expected for the treatment of tumors with cold phenotype, PD1 inhibition coupled with empty vector electroporation showed no antitumor potency (Fig. 7e–g), in agreement with the reports indicating favorable therapy outcomes in tumors with present neo-antigens and immuno-active TME[72,73]. However, PD1 inhibition in the NT GSDMD + IL-1β + IL-18-treated mice markedly improved the antitumor effects, leading to full remission of 50% of mice (Fig. 7e–g). In contrast, gene *Ctla4* encoding CTLA4 checkpoint was only modestly upregulated in several tumor samples (Fig. 6c). In line with those results, the combined therapy with anti-CTLA4 antibodies and NT GSDMD + IL-1β + IL-18 did not abrogate tumor growth (Fig. 7h–j).

We show that the abortive action of a combination of cytokines IL-1β and IL-18 on pyroptosis depends on IFN signaling and can be at least partially ameliorated by anti-PD1 ICI therapy.

### Cytokine-armed pyroptosis requires a functional immune system for tumor eradication

To define the importance of the immune system in mediating antitumor responses of cytokine-armed pyroptosis, we induced B16F10 tumor formation in immunocompromised B-NDG mice (mice with NOD scid background with IL-2-cytokine family dysfunctionality due to the deletion of the gene coding for the common gamma receptor chain (*Il2rg*)) that lack functional B, T and NK cells. We show that NT GSDMD electroporation does not lead to a decrease in tumor growth, not even when pyroptosis was enriched with IL-1β pretreatment (Fig. 8a–c and Supplementary Fig. 11b) and all animals were rapidly eliminated demonstrating that the protective effects of pyroptosis and cytokine-armed pyroptosis depend on the immune system.

The induction of immunogenic cancer cell death instigates migration of immune cells into the TME transforming immunologically cold tumors into hot tumors[18,48]. To observe the effect of NT GSDMD-induced pyroptosis and a combination with proinflammatory cytokine on the infiltration of CD8+ cells into TME, we performed laser ablation inductively coupled plasma mass spectrometry (LA-ICP-MS) on tumor tissue sections using metal-conjugated antibodies which enable spatially resolved analysis of tumor-infiltrating cells[74]. Predominantly

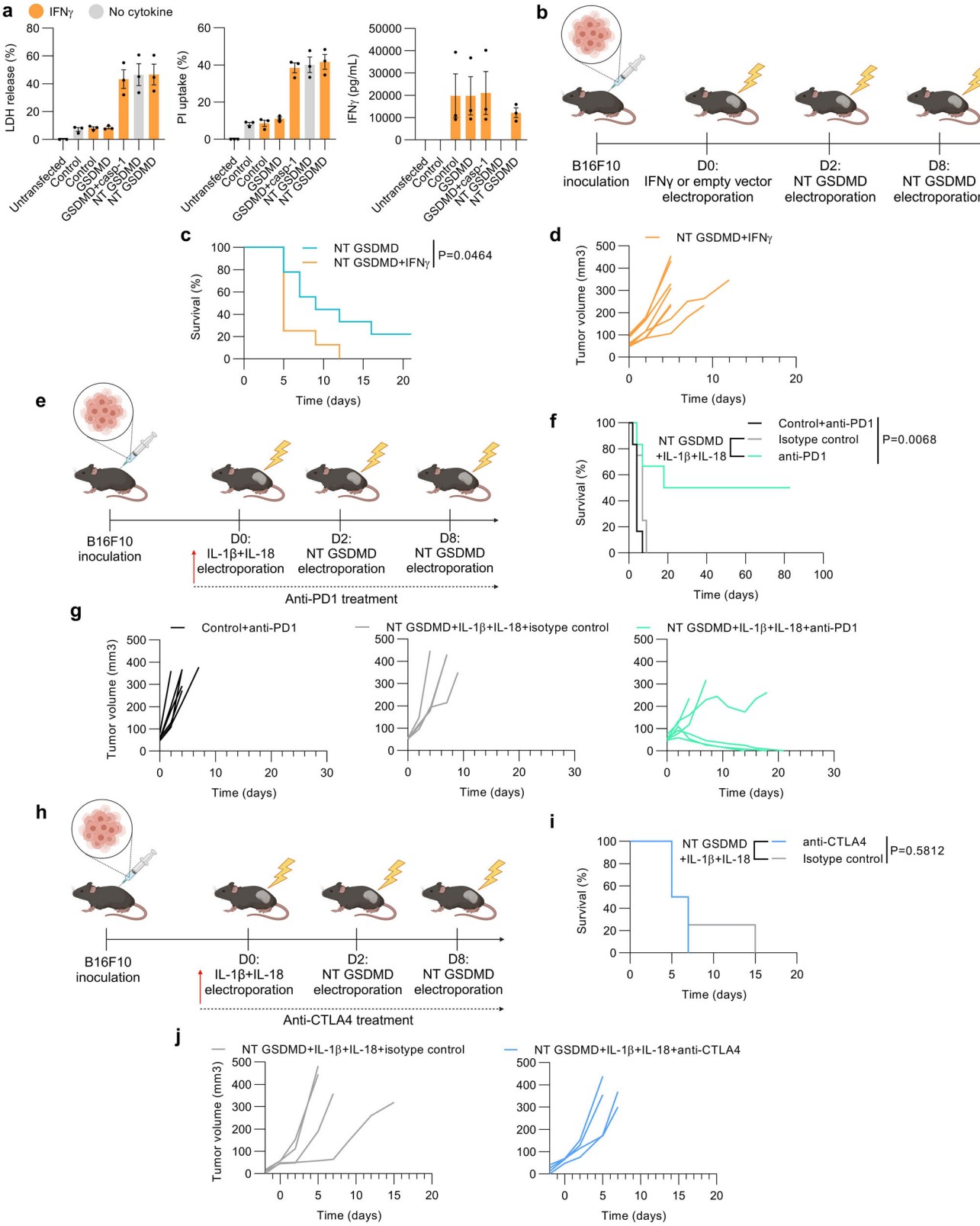

background labeling was observed for empty vector-treated tumors, while NT GSDMD treatment induced CD8+ cell infiltration that was additionally boosted with IL-1β pretreatment (Fig. 8d). Furthermore, we analyzed immune cell populations in tumors that underwent either control (empty vector electrotransfer treatment) or IL-1β-armed pyroptosis treatment. Compared to the control, cytokine-armed pyroptosis induced an intratumoral enrichment of NK cells and CD3-

positive subpopulations (T cells), including CD8+ cytotoxic T cells (Fig. 8e and Supplementary Fig. 11c).

To further validate the functional relevance of tumor-enriched populations, we performed IL-1β-enriched NT GSDMD treatment of B16F10 melanoma in immunocompetent C57BL/6 mice with depleted NK1.1 and CD8β-expressing cells. We found that depleting NK1.1-positive and CD8-positive cell populations abolished antitumor

**Fig. 7 | Infective NT GSDMD + IL-1β + IL-18 treatment is characterized by interferon signaling and the upregulation of immune checkpoints. a** In vitro screening of IFNγ cotransfected with GSDMD variants. HEK293T cells were transfected with IFNγ and GSDMD variants and LDH release, PI uptake, and ELISA assays were conducted. **b** Experimental scheme. Subcutaneously growing tumors were electroporated with 20 μg IFNγ (n = 8) or empty vector-coding plasmid followed by electroporation of an equal amount of NT GSDMD plasmid after 2 days (n = 9). The same NT GSDMD group as in Fig. 5c is shown. Kaplan-Meier curve (**c**) and tumor volume changes (**d**) present the NT GSDMD + IFNγ treatment efficacy compared to the NT GSDMD treatment. **e–j** Anti-PD1 but not anti-CTLA4 restores antitumorigenic activity of fully deconstructed inflammasome (NT GSDMD + IL-1β + IL-18). Mice received i.p. 200 μg anti-PD1 (n = 6) (**e**) or anti-CTLA4 (n = 4) antibodies (**h**) with appropriate isotype controls on the same day as IL-1β + IL-18 or empty vector control electroporation, and every 3rd day until remission or euthanization. Survival (**f**, **i**) and tumor volume (**g**, **j**) were documented over time. Results shown in (**a**) are mean ± SEM representative of three independent experiments. Time in (**c**, **d**, **f**, **g**, **i**, **j**) is determined as days after the treatment, and a log-rank test was conducted for survival data analysis (**c**, **f**, **i**). Source data are provided as a Source Data file. Figure 7b created in BioRender. Hafner Bratkovic, I. (2024) https://BioRender.com/z97l709; Fig. 7e created in BioRender. Hafner Bratkovic, I. (2024) https://BioRender.com/a77b958; Fig. 7h created in BioRender. Hafner Bratkovic, I. (2024) https://BioRender.com/n94v358.

response as all animals had to be rapidly euthanized (Supplementary Fig. 11d–f), whereas depleting only NK1.1 or CD8+ cells led to ~17% survival compared to the control IgG treatment where 50% of mice were alive at the end of the experiment (Supplementary Fig. 11d–f).

These results imply that cytokine-armed pyroptosis-induced tumor rejection requires functional NK and CD8+ cell-mediated responses and is in line with recent reports demonstrating that the antitumor properties of GSDMD-mediated pyroptosis largely depend on NK and T cell-mediated immunity[23,26].

### Cytokine-armed pyroptosis is tumor-agnostic and ignites systemic protective immunity

The ultimate goal of cancer immunotherapy is to induce long-lasting protective immunity. We tested if locally induced pyroptosis in the primary tumor is capable of inducing a protective response against the distant tumor. 31 days after the primary tumor inoculation surviving mice were rechallenged with B16F10 cancer cells administered into the other flank (Fig. 9a). Intratumoral therapy of primary tumors with cytokine-enriched pyroptosis generated robust protection against secondary tumors with all of the inoculated mice remaining tumor-free (Fig. 9b–d and Supplementary Fig. 12a), whereas all of the naïve mice injected with the same amount of B16F10 cells developed palpable tumors (Fig. 9e and Supplementary Fig. 12a). With this experiment, we showed that cytokine-enriched pyroptosis elicits systemic protective immunity.

In the final set of experiments, we explored if this approach triggers antitumor immunity and cancer cell eradication in different types of tumors. 4T1 mammary carcinoma is another model with a cold-tumor phenotype and poor response to immunotherapies[75]. 4T1 tumors implanted in syngeneic mice were treated by the same protocol as for the melanoma model (Fig. 9f). In the NT GSDMD-treated group we achieved prolonged survival compared to the empty vector-treated mice, yet tumor growth did not cease (Fig. 9g, h), whereas NT GSDMD + IL-1β therapy resulted in tumor suppression and survival of 17% of treated mice (Fig. 9g, h and Supplementary Fig. 12b, c). These results demonstrate that cytokine-armed pyroptosis has the potential to eliminate immune desert tumors of different origins.

In contrast to cold tumors, inflamed tumors are characterized by lymphocyte infiltration, but the TME suppresses the action of effector cells, partially by upregulation of immune checkpoints[3]. We were interested also in how deconstructed inflammasomes perform in immunologically hot tumors. We selected allogeneic subcutaneous CT26 colorectal carcinoma model[75,76] (Supplementary Fig. 13a). While empty vector-electroporated tumors grew rapidly, the IL-1β-enriched pyroptosis efficiently aborted tumor growth and led to tumor elimination and long-lasting survival in 80% of mice (Supplementary Fig. 13b–d). With that, we demonstrate that deconstructed inflammasomes can be used also for the treatment of immune cell-infiltrated tumors.

Our results emphasize the potential of cytokine-armed pyroptosis for induction of systemic antitumor immunity and demonstrate that this treatment strategy is effective in the treatment of tumors of different origins and immunotypes.

## Discussion

Solid tumors evolved diverse strategies to prevent durable immune responses. To overcome tumor-mediated immunosuppression various approaches are being validated with combinatory therapies employing multiple lanes of immune signaling performing better than single-axis-oriented treatments[77,78]. Motivated by the inflammasomes, nature's "combinatory therapy", we designed an armed pyroptosis approach that couples GSDMD-induced ICD with immunostimulatory cytokines to provide long-lasting immunity against diverse types of cancer.

Different cell death pathways are being explored for the induction of cancer cell ICD as anti-cancer treatment[18,79]. Intratumor delivery of mRNA encoding necroptosis executor MLKL was shown to affect primary tumor growth as well as to protect against distal and disseminated tumor formations in mouse cancer models[80]. Another example of necroptosis-inducing antitumor immunity was demonstrated via the RIPK3 induction system[81]. In addition to immunogenic apoptosis and necroptosis, but not ferroptosis[42], pyroptosis-mediated tumor suppression was observed with nanoparticle-delivered chemically cleavable GSDMA3[26], NT GSDMD-transduced tumor cells[82] and AAV-delivered NT GSDMD[27,83]. We developed a toolbox of engineered GSDMD variants that can induce pyroptosis spontaneously when introduced into tumors or on-demand, controlled by a non-toxic small molecule. These systems independently of endogenous inflammasome components enable pyroptosis induction with different levels of efficiency and regulation for future integration into more complex systems[45,84]. When applied to relatively large tumors, the induction of ICD with GSDMD variants proved to be potent in B16F10 melanoma tumor surveillance with around 20–25% long-term survivors regardless of whether NT GSDMD was introduced (Fig. 2) or GSDMD cleavage was induced by the addition of small molecule (Fig. 3). Our study corroborates previous studies that showed that a low level of cancer cell pyroptosis is sufficient for antitumor action[26,85] as the treatment engages the immune system to eradicate the tumor[23,26].

Pyroptotic cell death represents only one axis of inflammasome signaling. The release of DAMPs by dying cells was insufficient for CD8+ T cell cross-priming in the case of necroptosis or inflammatory apoptosis and NF-κB-dependent cytokine expression and production of inflammatory cytokines by the dying cells is needed to engage antitumor immunity[41,42]. To refine ICD-contingent immunotherapy we designed combinatory treatment utilizing inflammasome-dependent cytokines IL-1β or IL-18, or cytokine IL-12 in combination with cell death-causing NT GSDMD, which represents a deconstruction of two key inflammasome functions. Previous studies demonstrated that IL-1β and IL-18 are crucial inflammasome-driven mediators of antitumor defenses. IL-1β is a potent inflammatory cytokine affecting dendritic cells (DCs), NK, and T cells[30,51,86,87], whereas IL-18, with its ability to activate NK and T cells, has shown promise in preclinical cancer immunotherapy studies[34,35]. IL-12 is another powerhouse of antitumor immunity and was utilized to empower CAR T cell immunotherapy[38]. As self-replicating IL-12 RNA encapsulated in lipid nanoparticles, it was used in combination with ICI therapy, as well as oncolytic viruses[39,40], however, dose-related adverse side effects limited IL-12 treatments[53]. We show that single

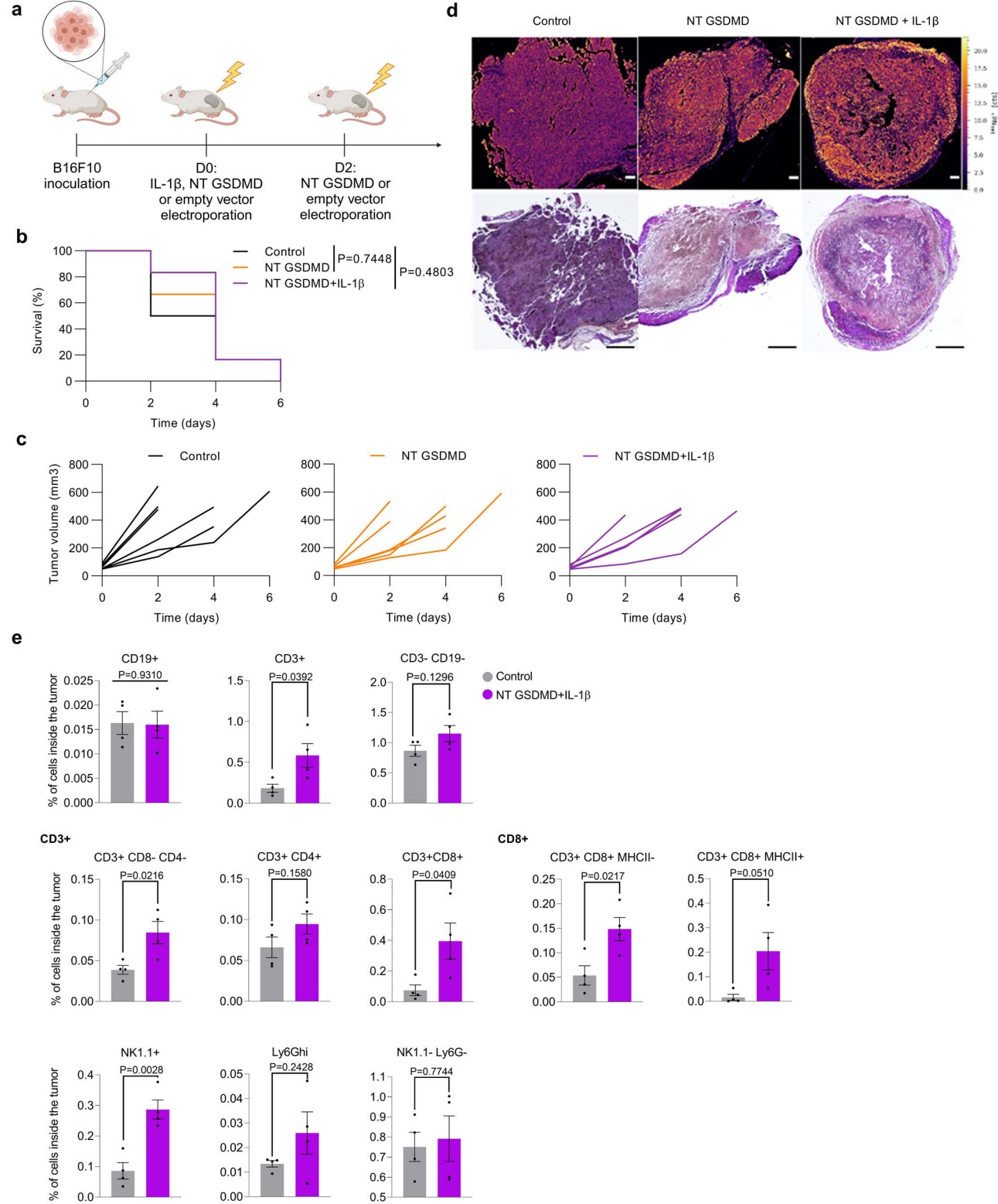

**Fig. 8 | Antitumor properties of GSDMD-mediated pyroptosis depend on the immune system. a** Schematic representation of the experiment in the immunodeficient mice. B-NDG mice were inoculated with B16F10 cells and treated with IL-1β, NT GSDMD, or empty vector plasmids (20 μg) (*n* = 6). Kaplan-Meier curves (**b**) and tumor volume charts (**c**) outline the pervasiveness of the melanoma tumor in immunodeficient mice. **d** LA-ICP-MS images (top) and Hematoxylin-eosin stain (bottom) of tumor tissue sections. Scale bars 100 μm (LA-ICP-MS) and 400 μm

(HE). **e** Analysis of immune cell populations in tumors shown as a % of cells inside the tumor. Time in days succeeding the first treatment is shown (**b, c**) and survival chart (**b**) was analyzed using a log-rank test. Plots (**e**) are presented as mean ± SEM of four different tumors per group and comparisons were calculated by unpaired two-tailed *t*-test. Welch's correction was used in case of different variances. Source data are provided as a Source Data file. Figure 8a created in BioRender. Hafner Bratkovic, I. (2024) https://BioRender.com/z11b682.

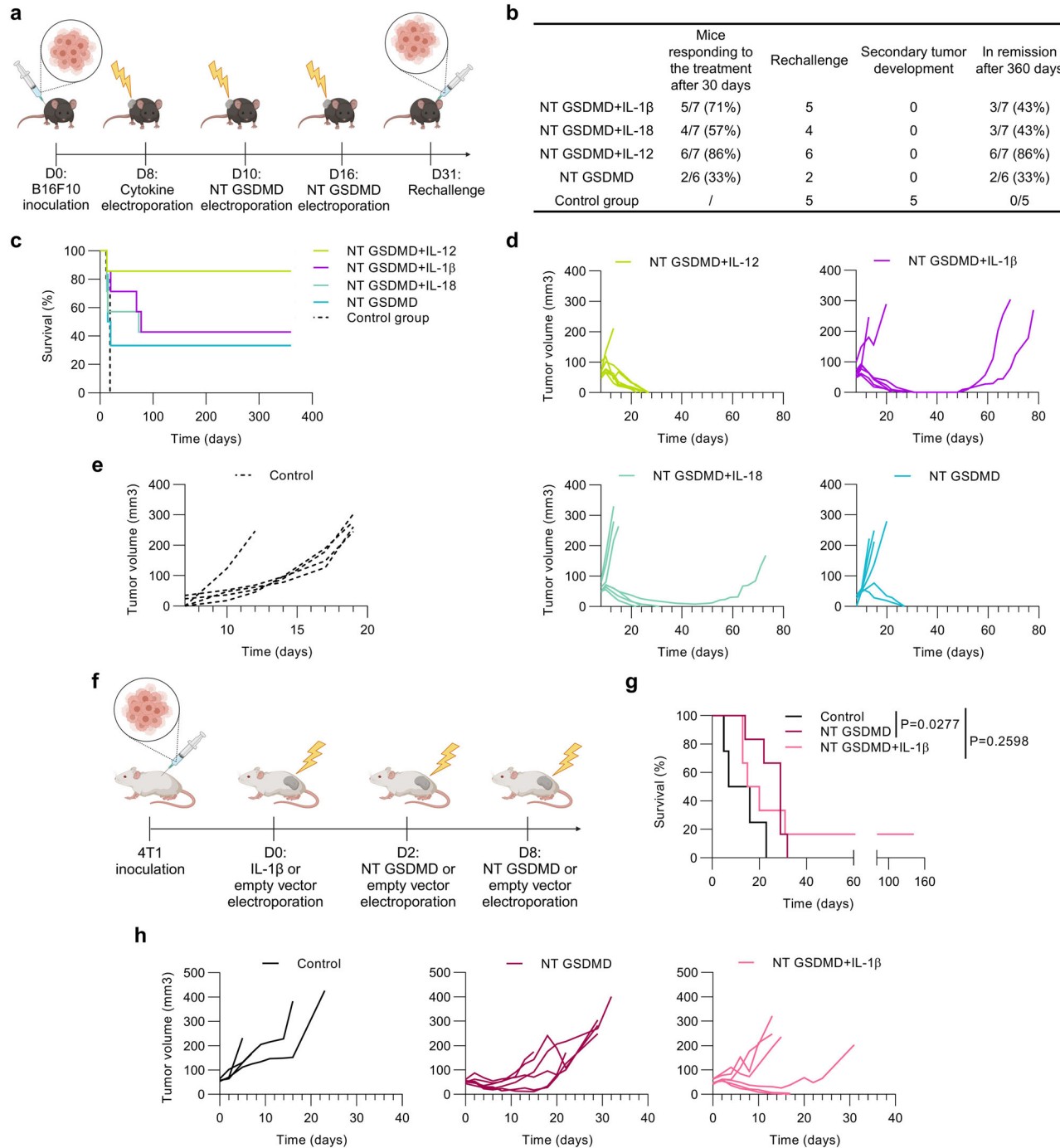

**Fig. 9 | Cytokine-armed pyroptosis drives immune response against tumor rechallenge. a** Timeline of rechallenge experiment. B16F10 tumor cells were s.c. injected into the right flank of the animals and treated following the same protocol as previously described ($n = 7$ for each cytokine and $n = 6$ for NT GSDMD alone). 31 days after the primary tumor inoculation, surviving mice were rechallenged with the same amount of B16F10 cells s.c. administered into the left flank. As a control, naïve mice ($n = 5$) received an equal amount of tumor cells. **b** Tabular display of survival data. **c** Survival curves analyzed by log-rank test. Tumor growth curves of primary (treated) tumor (**d**) or tumor injected into naïve mice at the same time as the second injection of B16F10 melanoma cells (**e**). Time is defined as post-B16F10 injection for primary or secondary tumors (**c–e**). **f–h** 4T1 tumor model. **f** Experimental scheme. BALB/c mice were s.c. implemented with 4T1 cancer cells and when tumors reached 50 mm³, they were electroporated with 20 µg of plasmid coding for IL-1β ($n = 6$) 2 days prior to NT GSDMD ($n = 6$) electrotransfer (20 µg). 20 µg of empty vector electroporation was used as a control ($n = 4$). Representative Kaplan-Meier (**g**) and tumor volume plots (**h**), where the timeline following the first treatment is shown. Survival data (**g**) were analyzed with a log-rank test. Source data are provided as a Source Data file. Figure 9a was created in BioRender. Hafner Bratkovic, I. (2024) https://BioRender.com/d24b373; Fig. 9f created in BioRender. Hafner Bratkovic, I. (2024) https://BioRender.com/p33z765.

electroporation of plasmids encoding cytokines did not induce systemic inflammation, but was also insufficient to drive potent antitumor immunity which corroborates previous preclinical studies[55,56] and is in line with the protocols in the clinical studies that used several rounds of electroporation of the IL-12-encoding plasmid[88]. On the other hand, intratumorally expressed cytokines synergized with delayed ICD in the potentiation of antitumor responses.

Based on dissimilar signaling pathways employed by those cytokines and targeted cells[89] it is not surprising that the whole tumor

transcriptome analysis demonstrates high diversity in enriched pathways of tumor-eradicating treatments. In line with studies demonstrating tumor-suppressing activities of selected cytokines and of immunogenic cell death induction[23,26], we show that armed pyroptosis ameliorated tumor surveillance by promoting CD8+ T cell and NK1.1+ cell infiltration into the TME and that tumor suppression relies on CD8+ and NK cell-mediated immunity since their depletion antagonized antitumor responses. Further studies are needed to identify the interplay of immune cells in supporting tumor clearance. Our study confirms that arming pyroptosis with cytokines that initiate inflammatory cascades, is pivotal for a coordinated inflammation and orchestration of adaptive immunity against cancer.

There are several rationales why designing synthetic inflammasome-inspired systems instead of stimulating the endogenous inflammasomes. The first is that cancer cells suppress cell death pathways, including the expression of inflammasome pathway and gasdermins[21–25]. Further, there are contradictory reports regarding the involvement of endogenous inflammasomes in tumor propagation and antitumor defense (reviewed in refs. 90–94). Chemotherapy was shown to boost antitumor immunity by inducing IL-1β through PTEN-primed NLRP3 inflammasome activation in myeloid cells[95,96]. However, several chemotherapeutics engage NLRP3 inflammasome in myeloid-derived suppressor cells, thus antagonizing the effect of chemotherapy[97,98]. IL-1β and IL-18 can differently promote tumor progression, e.g. by aggravating chronic inflammation, promoting cancer cell proliferation, angiogenesis, and tumor evasion from immune surveillance, but described protumorigenic actions depend on many variables such as tumor type, expression characteristics, and cell source of proinflammatory cytokine[92,99–101]. A group of oxidized phospholipids that promote DC hyperactive state with prolonged secretion of IL-1β in the absence of cell death[102,103] facilitate superior antitumor responses in comparison to canonical NLRP3 inflammasome triggers, such as vaccine adjuvant alum that induces DC pyroptosis[102,104]. Our study does not negate the potential of inhibition of chronic inflammasome signaling[105] with several clinical trials ongoing (reviewed in refs. 92,93,99). Instead, we demonstrate the therapeutic potential of a transient deconstruction of inflammasomes in tumor cells with either IL-1β or IL-18 in combination with pyroptosis. However, even transiently deconstructed inflammasome employing both inflammasome-dependent cytokines and pyroptosis exhibited no protective effects. Remarkably, the whole tumor transcriptome analysis revealed that IL-12 enriched pyroptosis and NT GSDMD + IL-1β + IL-18 treatments with different therapeutic outcomes exhibit enrichment in interferon signaling. This phenotype is surprising for ineffective treatment, as a loss of interferon signaling and consecutive presentation of MHC I class molecules is one of the common traits in immune evasion[57–64]. Recent studies connected IFNγ signaling to immunosuppression of NK and CD8+ T cell immunity through upregulation of classical and nonclassical MHC I molecules[67,106], and upregulation of immune checkpoints[71]. We demonstrated that observed immunosuppression was, at least in part, dependent on the expression of immune checkpoints, as anti-PD1 treatment, when applied with the NT GSDMD + IL-1β + IL-18 therapy, restored the antitumor immunity. Furthermore, cytokine-armed pyroptosis utilizing IFNγ directly had no tumor surveillance capacity, supporting the positive correlation between loss of tumor IFNγ signaling and tumor eradication by the immune system[67]. Our study suggests that sometimes "less is more" and that modulation of antitumor immunity requires a precise balance between immunostimulatory signals and the kinetics of their resolution.

In contrast to previously used delivery methods, we utilized electrogenic transfer of plasmid DNA, which can be efficiently produced in bacterial cells, while the production of AAV vectors encoding toxic proteins is substantially more challenging[27]. Plasmid DNA electroporation has been explored in cancer clinical trials for the introduction of cytokines[107,108]. As we used this system for comparison of different treatments, we did not optimize the treatment regimen, thus it is highly likely that the survival rate could be even increased if started earlier when the tumors are smaller and with more frequent electroporations and also using different electroporation equipment such as multi-pin electrodes. This is particularly true for the 4T1 breast cancer model, where Wang and coworkers demonstrated that a single round of treatment with GSDMA3 nanoparticles only slightly delays tumor growth[26], which is in line with our results. In addition to electroporation of accessible tumors, tumors or surgical margins can be electroporated during tumor resection surgery, or special electrodes that enable deep tissue electroporation can be used to treat deep-seated tumors[109]. By developing different GSDMD-based pyroptosis mediators we expanded the portfolio of ICD inducers that could be incorporated into different delivery systems, which can be useful, particularly for less accessible tumors. The size of our largest ICD inducer, a polycistronic dDmrB casp-1:GSDMD construct, is 3135 bp, which can easily fit into delivery systems such as AAVs[110]. As the armed pyroptosis approach was designed to be modular, we can foresee that the platform could evolve in different directions, particularly it will be interesting to test the approach together with ICI therapy in cold tumors, especially since several preclinical studies have shown that induction of ICD improves the effect of ICI therapy[26,27,80,85].

We present an innovative therapeutic approach in which the cytokine component combined with delayed cancer cell pyroptosis drives potent and persistent antitumor responses without adverse systemic side effects. The designed approach does not rely on the expression of cell death effectors or the presence of cells that could provide immunostimulatory signals, which is demonstrated by its pleiotropic activity in diverse types of solid tumors, including difficult-to-treat immune desert tumors. Novel designs including combination with ICI therapy and diverse modes of delivery could further enhance its translational potential in the treatment of non-accessible tumors and pave the way for the clinical application of combinatory therapy fusing immunostimulatory components and cancer cell death.

## Methods

### Study approval
All animal experiments were performed according to the 3R principles and directives of the EU 2010/63 and were approved by the Administration of the Republic of Slovenia for Food Safety, Veterinary, and Plant Protection of the Ministry of Agriculture, Forestry, and Foods (Permit Number U34401-9/2020/9, U34401-24/2023/4 and U34401-18/2023/8).

### Construct preparation
All used constructs were cloned into the pcDNA3 vector (Invitrogen). Mouse WT GSDMD gene was ordered from Sino Biological (MG5A2835-U). NT I105N GSDMD and GSDMD[TEV] variants were generated by introducing point mutation and TEV cleavage site via site-directed mutagenesis, respectively, using a Phusion HF polymerase (Thermo Fisher Scientific). Mouse IL-18 (Sino Biological, MG50073-M) was amplified without the pro-part using primers, HA-tag-encoding sequence was added to the 3′ and inserted into the pcDNA3 vector with Gibson assembly protocol. Mouse IL-1β was synthesized as G-blocks (IDT) without the pro-part and with the addition of HA-tag on the C-terminus. Mouse IFNγ (Sino Biological, MG50709-M) and IL-12 p40p35 (synthesized as G-blocks (IDT)) were amplified using primers and Gibson assembly was used to clone it into the pcDNA3. dDmrB casp-1ΔCARD was prepared previously[43]. GSDMD:T2A:dDmrB casp-1 and dDmrB casp-1:T2A:GSDMD were prepared using Gibson assembly. The preparation of plasmids encoding FKBP cTEV and FRB nTEV is described in ref. 45. Construct amino acid sequences can be found in Supplementary Table 1 and oligonucleotide sequences in Supplementary Table 2.

## Cell lines and cell culture conditions

HEK293T cells were obtained from the American Type Culture Collection (ATCC, CRL-3216). Bioware® Brite Cell Line B16F10 Red-FLuc cells were ordered from PerkinElmer (cat. no. BW124734). Cells were maintained in Dulbecco's Modified Eagle's Medium (DMEM) + GlutaMax (Gibco, Life Technologies), supplemented with 10% Fetal Bovine Serum (FBS) (Gibco, Life Technologies) at 37 °C and 5% $CO_2$. Murine BALB/c colorectal carcinoma cell line CT26 and murine BALB/c breast cancer cell line 4T1 cells were a kind gift from Prof. M. Čemažar (Institute of Oncology, Ljubljana, Slovenia) and were maintained in Roswell Park Memorial Institute 1640 medium (RPMI) (Gibco, Life Technologies) supplemented with 10% FBS at 37 °C and 5% $CO_2$. The generation of immortalized bone marrow-derived macrophages (iBMDM) from wild-type and GSDMD-KO mice was previously described[43]. All cell lines were tested mycoplasma-negative and were authenticated by morphology only.

CT26 Red-FLuc and 4T1 Red-FLuc were generated by lentivirus transduction. For lentivirus production, HEK293T cells were seeded at $4 \times 10^6$ cells/10 cm-plate. 24 h later, cells were transfected with 15 μg of LVX vector expressing FLuc, 10 μg of VSV-G-encoding vector (Cell Biolabs), and 5 μg of psPAX2 vector (Addgene, 12260, a gift from Didier Trono) using jetPEI. After 48 h, cell supernatant containing viral particles was filtered through a 0.45 μm filter (Sartorius) and ultracentrifuged at 4 °C for 2 h/100.000 × g (Beckman Coulter). The viral particle pellet was resuspended in PBS.

For 4T1 Red-FLuc and CT26 Red-FLuc generation, cells were seeded at $1 \times 10^5$ cells/well in a 12-well plate. The next day, they were transduced with lentiviruses in the presence of 8 μg/ml polybrene (Invivogen). Seven days later, FLuc-positive cells were selected using puromycin (0.5 mg/ml) (Invitrogen). The presence of FLuc was confirmed by bioluminescence imaging using IVIS Lumina Series III (Perkin Elmer).

## Transfection of HEK293T and B16F10 cells

HEK293T cells were counted using a Luna-II cell counter (Logos Biosystems) and plated at $2.2 \times 10^4$ cells/well in a black 96-well plate with an optically clear bottom. After 24 h transfection with plasmids coding for different GSDMD variants, cytokines, casp-1, TEVp, or empty vector was performed using 12 μl of in-house prepared Polyethylenimine (PEI) cellular transfection reagent per 1000 ng of DNA.

B16F10 cells were seeded at $1 \times 10^4$ cells/well in a black 96-well plate. The next day, cells were transfected with a transfection mixture composed of 100 ng DNA with 0.1 μL JetOptimus (Polyplus)/well. 6 h later, the medium was exchanged.

## Propidium iodine uptake

To measure cell membrane permeability, the cells were incubated for 30 min with a diluted PI solution with a final concentration of 3.33 μg/ml. The plate was spun down at 500 × g for 5 min and measured using the multiple reader SinergyMx (BioTek) and Gen 5.1.10 software (BioTek) with program settings: bottom reading of fluorescence with an excitation wavelength of 530 nm and emission wavelength of 617 nm. The percentage of the PI uptake was calculated using the untreated cells as negative control and lysis buffer-treated cells as 100% PI uptake.

## LDH cytotoxicity assay

LDH release in supernatants was measured using CyQuant LDH cytotoxicity assay (Invitrogen) according to the manufacturer's protocol. The percentage of the LDH release was calculated using the supernatant of the untreated cells as negative control and lysis buffer-treated cell supernatant as 100% LDH release.

## Visualization of pyroptotic cells by confocal microscopy

$3 \times 10^4$ B16F10 cells were seeded into an 8-well ibidi slide. The next day cells were transfected with 300 ng DNA and 0.3 μL JetOptimus (Polyplus)/well and 6 h later, the medium was exchanged. After 24 h staining with Hoechst 33342 (Thermo Scientific) and PI (Thermo Scientific) was conducted and cells were imaged using the TCS SP5 confocal microscope (Leica).

## Mice

As described at the beginning of the Methods Section, all animal experiments were performed following EU and National Law and approved by the Administration of the Republic of Slovenia for Food Safety, Veterinary, and Plant Protection of the Ministry of Agriculture, Forestry, and Foods. Eight to twelve-week-old mice were used for the experiments. Laboratory mice C57BL/6J OlaHsd, BALB/c OlaHsd, and B-NDG were housed in IVC cages GM500 (Techniplast), fed standard chow (Mucedola) and tap water was provided ad libitum. The cages were enriched using Nestlets nesting material and mouse houses. Mice were maintained in 12–12 h dark-light cycle at ~40–60% relative humidity with ambient temperature (22 °C). All animals used in the study were healthy and accompanied by a health certificate from the animal vendor. Both male and female mice were used in the study. Health and microbiological status was confirmed by FELASA recommended Mouse Vivum immunocompetent panel (QM Diagnostics). All experiments were conducted during the light cycle.

## Tumor electroporation

After injection of plasmid DNA in 50 μl PBS in several positions of a tumor, tumors were electroporated using Gene Pulser Xcell Total Electroporation System (Bio-Rad Laboratories) with the parameters: Voltage 900, pulse width 100 μs, pulse interval 1 s, and pulse number 5. Electrodes were clasped on the tumor and the first pulse was initiated, then the electrodes were moved into another position on the tumor and the second pulse followed (as depicted in Supplementary Fig. 3a), and this was repeated until 5 pulses were reached. For each experiment and mouse model, the extensive treatment timeline is described in the sections below.

## Melanoma tumor model

For the tumor challenge experiment, $2 \times 10^6$ B16F10 Red-FLuc cells were subcutaneously (s.c.) inoculated into the right flank of 8–12 weeks old C57BL/6J OlaHsd mice. When tumors were palpable and reached ~50 mm³, determined using the ellipsoid volume formula (width × length × height × π)/6, 20 μg of pcDNA3 empty vector, IL-1β, IL-18, single chain IL-12 or IFNγ plasmid were injected into the tumor followed by electroporation as described above. No electroporation was conducted for the naked uptake (gymnosis) of the plasmids. When the combination of IL-1β and IL-18 was used, 10 μg of each plasmid was electroporated. For half-dose IL-1β, 10 μg of IL-1β-encoding plasmid and 10 μg of pcDNA3 empty vector were electroporated. Two and eight days after the first electroporation or as depicted in the experimental scheme in the figures, tumors were injected with 20 μg of pcDNA3 empty vector or NT GSDMD plasmid and electroporated using the same parameters. The tumor volume was measured three times a week with an electronic digital caliper and mice were humanely euthanized when the length, width, or height of the tumor exceeded 12 mm, which is a defined humane endpoint, or when the tumor showed signs of ulceration. For the experiment using the inducible system, 20 μg of the dDmrB casp-1:GSDMD plasmid was injected and electroporated as described above. One day after the electroporation, dimerization was induced by i.t. administration of 30 μl 5 μM AP20187 (MedChemExpress) dimerizer. Mice in the control group were injected with 30 μl PBS. For bulk RNA sequencing analysis of the tumor microenvironment, mice were euthanized 14 days after the inoculation of tumor cells (6 days after the first electroporation treatment), and tumors were collected for further analysis. Half of the tumor was dedicated for bulk RNA sequencing and the other half was fixed overnight with 10%

neutral buffered formalin (Sigma-Aldrich). Additionally, blood samples of treated mice were collected for serum isolation.

For the experiment with immunodeficient mice, $2 \times 10^6$ B16F10 Red-FLuc cells were s.c. inoculated into the right flank of 8–12 weeks old B-NDG mice (NOD.CB17-*Prkdc*$^{scid}$ *IL2rg*$^{tm1}$/BcgenHsd). When mice developed ~50 mm$^3$ tumors the same protocol for pcDNA3 empty vector, IL-1β, or NT GSDMD plasmid injection and electroporation was followed as described above. Tumor volume was sized three times a week and mice were humanely euthanized when the length, width, or height of the tumor exceeded 12 mm.

### 4T1 and CT26 tumor models
$2 \times 10^6$ 4T1 Red-FLuc or CT26 Red-FLuc were s.c. injected into the right flank of 8–12 weeks BALB/c OlaHsd or C57BL/6 J OlaHsd mice, respectively. When tumors reached ~50 mm$^3$, calculated with formula (width × length × height × π)/6, 20 µg of plasmids (empty vector or IL-1β) were i.t. injected and electroporated using the same settings as for the B16F10 tumor model. Two and eight days after the first treatment palpations were injected with 20 µg plasmid encoding NT GSDMD or 20 µg empty vector followed by electroporation. Tumor growth was monitored three times a week and when the tumor's length, width, or height exceeded 12 mm, mice were humanely euthanized.

### Rechallenge experiment
The primary tumor was inoculated and treated as described above. 15 days after the last NT GSDMD electroporation of the primary tumor, surviving mice were s.c. implanted with $2 \times 10^6$ B16F10 Red-FLuc cells into the left flank. Primary and secondary tumors were monitored three times a week and mice were excluded from the experiment when the length, width, or height of the primary or secondary tumor exceeded 12 mm.

### In vivo depletion of NK1.1/CD8b2-positive cells
CD8+ and/or NK cells were depleted by i.p. injections of 200 µg of anti-mouse CD8b.2 monoclonal antibody (clone 53-5.8, product no. C2832, Leinco Technologies) and/or anti-mouse NK1.1 monoclonal antibody (clone PK136, product no. N123, Leinco Technologies), respectively, 1 day before and after initiation of therapy, and again every 6 days thereafter until mice were euthanized or no palpable tumor was detected. As a control for depletion, mouse IgG2a (clone C1.18.4, product no. I-118, Leinco Technologies) was used. All antibodies were prepared in 100 µl sterile PBS.

### PD1 and CTLA4 blocking antibodies
Mice were administered i.p. 200 µg of PD1 blocking antibodies (anti-mouse CD279, clone RMP1-14, product no. P362, Leinco Technologies) or rat IgG2a (clone 1-1, product no. I-1177, Leinco Technologies) for the isotype control, or CTLA4 blocking antibodies (anti-mouse CTLA4, clone 9D9, C2855, Leinco Technologies) or mouse IgG2b isotype control (clone MCP-11, product no. I-119, Leinco Technologies), the same day as the first treatment and again every 3 days until mice were euthanized or no palpable tumor was detected.

### In vivo bioluminescence imaging
Tumor progression was monitored weekly by IVIS Lumina Series III (Perkin Elmer) after the s.c. injection of 150 mg D-Luciferin per kg body weight (Xenogen). Data were analyzed using Living Image 4.7.3 (PerkinElmer) and the luminescent signal was quantified as average radiance (p/s/cm²/sr).

### In vivo propidium iodide staining of tumor cell pyroptosis
B16F10 tumor-bearing mice were electroporated with 20 µg of pcDNA3 empty vector or NT GSDMD plasmid and after 24 h sedated using isoflurane (Sedaconda inhalation vapor, Sedana Medical) and i.t. administered with 2.5 mg/kg of propidium iodide for labeling of

pyroptotic cells in vivo. After 20 min mice were euthanized, and tumors were collected and flash-frozen in liquid nitrogen. Tumors were placed into OCT-containing Cryomold molds and 7 µm-thick cryosections were prepared using Routine Cryostat MEV (Slee). Cryosections were thawed at room temperature and washed in PBS. Next, the samples were stained with Hoechst 33342 (Thermo Scientific) for 15 min at room temperature and again washed in PBS. Imaging was performed using the TCS SP5 confocal microscope (Leica) and fluorescence-based quantification of PI-stained cells was carried out using the ImageJ software (National Institutes of Health, Bethesda, USA).

### Tissue staining
After 48 h of fixation in 10% neutral buffered formalin (Sigma-Aldrich) and gradient ethanol dehydration, the tumor samples were embedded in paraffin (Leica Biosystems), sectioned into 7 µm pieces using Microtome (Leica Biosystems) and mounted on adhesive coated slides (Leica Microsystems). After deparaffinization and hydration, sections were stained with Mayer's Hematoxylin Solution (Merck) and Eosin Y-solution (Sigma-Aldrich), or Trichrome stain (Masson) kit (Sigma-Aldrich) according to the manufacturer's instructions.

### Primary cell cultures
Primary cultures of B16F10 melanoma were established by cutting the tumors into ~1 mm$^3$ pieces and culturing them as explant cultures in DMEM supplemented with 10% FBS and 100 U/ml penicillin–streptomycin (Gibco) at 37 °C and 5% $CO_2$. After 7–10 days, the cells were harvested and seeded for further experiments. After 24 h in culture (at passage 1), the cells were stained with Hoechst 33342 (Thermo Scientific) and imaged using the TCS SP5 confocal microscope (Leica).

### Cell proliferation assay
Primary cell cultures were seeded at a seeding density of $5 \times 10^3$ cells/well or $1 \times 10^4$ cells/well in a 48-well plate and cultured for 24–72 h. At selected time points the cells were fixed with 4% formaldehyde (w/v) and stained with crystal violet (Sigma), followed by bright field imaging (10× magnification; CellInsight CX7, Thermo Scientific). The area covered with cells was determined using ImageJ software (National Institutes of Health, Bethesda, USA), measuring this area as a percentage of the total field view area.

### Migration assay
Primary cell cultures were seeded at a seeding density of $5 \times 10^4$ cells/well in a 48-well plate and grown to confluence. The cell-free gap was formed by a 200 µl pipette tip followed by bright field imaging (4× magnification; CellInsight CX7, Thermo Scientific) at time 0 and after 24 and 48 h. The percentage of scratch closure was determined using ImageJ software (National Institutes of Health, Bethesda, USA).

### Systemic effects of cytokine-armed pyroptosis treatment
Blood was collected from the tail vein on days −4, 0, and 4, according to the first treatment, in multivette tubes containing a clotting activator (Sarstedt), spun down for 20 min at 3000 rpm and 4 °C, and ELISA was used for measuring the concentration of mouse IL-18, IL-1β, IL-12, IL-6, and IFNγ (Invitrogen) in mice sera according to the manufacturer's instructions. Replacement fluid (sterile 0.9% saline) was administered i.p. to the mice.

### Biochemical blood analyses
Terminal blood was collected from the mice and spun down for 20 min at 3000 rpm and 4 °C. Sera were analyzed using a VetScan Comprehensive Diagnostic Profile kit (Abaxis) with a VetScan VS2 analyzer (Abaxis) according to the manufacturer's instructions.

## Western blot

Cells were seeded at $1 \times 10^6$ cells/ml in a 12-well plate and lysed after 24 h. The protein concentration in the cell lysate was measured with BCA, proteins were then separated on SDS-PAGE gel and blotted onto the nitrocellulose membrane using iBlot 2 stacks and iBlot 2 Dry Blotting System (Invitrogen) or wet blotting system (Biorad). Primary anti-GSDMD antibodies (Abcam, ab209845, 1:1000), anti-caspase-1 p20 antibody (Casper-1, Adipogen, AG-20B-0042-C100, 1:1000), anti-ASC antibody (AL177, Adipogen, AG-25B-0006, 1:1000), anti-NLRP3 antibodies (Cryo2, Adipogen, AG-20B-0014-C100, 1:1000), anti-β-Actin (Cell Signaling Technology, 8H10D10, (#3700, 1:5000), α/β-Tubulin antibody Cell Signaling Technology (#2148, 1:5000) and secondary HRP-conjugated anti-rabbit (Jackson ImmunoResearch, 111-035-003, 1:3000) or secondary HRP-conjugated anti-mouse (Jackson ImmunoResearch, 115-035-003, 1:3000) were used for the detection of GSDMD and inflammasome components and housekeeping gene-coded proteins. SuperSignal West Pico or Femto Chemiluminescent Substrate (Thermo Scientific) were used for the detection of HRP-labeled bands with G-box (SynGene) and Genesnap 7.09 software (SynGene). Uncropped images are presented in Source Data file and in Supplementary Fig. 16.

## RNA isolation and bulk RNA seq

Total RNA was extracted from the fresh tumors using TriPure Isolation Reagent (Roche) according to the manufacturer's instructions. Collected RNA from the animals of the same group was pulled together and sent for bulk RNA sequencing in triplicates. Azenta Life Sciences (Genewiz, Leipzig) performed sample quality control, library preparation, and rRNA removal by polyA selection for mRNA species. The libraries were run on Illumina® NovaSeqTM, using a 2×150bp paired-end sequencing protocol, with a sequencing depth of 20–30 million reads per sample.

Data was analyzed using various tools within the RNAlysis software[111]. First, we performed QC, using FastQCheck and then paired-end adapter trimming, using CutAdapt[112]. Then we identified transcripts by using Kallisto[113] tool for pseudoalignment of trimmed samples to the mouse genome, utilizing pre-produced index and gtf files, downloaded from Kallisto transcriptome indices data source (version June 22 2019; kallisto 0.45.1, Ensembl v96 transcriptomes; https://github.com/pachterlab/kallisto-transcriptome-indices/releases; accessed Sep 29, 2023).

For differential expression and relevant statistical analysis, we used RStudio (RStudio 2023.09.1 + 494), specifically R-package DESeq2[114]. Data were normalized with an in-built function (median-of-ratios method), the threshold of significance was set to adjusted $P$-value $\leq 0.1$. If not stated otherwise, ggplot2 package was used to draw plots. Heatmaps were drawn with GraphPad Prism 8, with Z-scores, calculated from DESeq normalized data. For enrichment analysis we used significantly upregulated genes of each sample (adjusted $P$-value $< 0.1$ and $\log_2 FC > 0.5$) as an input to the online-available WEB-based Gene Set Analysis toolkit (WebGestalt) and performed Over-Representation Analysis (ORA), choosing functional database gene ontology (non-redundant biological process) and the reference set (mouse) genome[115].

## LA-ICP-MS

Deparaffinized tumor sections were incubated in antigen retrieval solution for 20 min at 75 °C and blocked with 3% BSA (Sigma-Aldrich) in DPBS (Gibco, Life Technologies) for 45 min at room temperature. Tumor sections were stained overnight at 4 °C with anti-mouse CD8a (53-6.7)-146Nd (Standard BioTools, 3146003B) (50× dilution). The instrumental setup used for LA-ICP-MS measurements was comprised of a laser ablation system (193 nm ArF* excimer; Analyte G2 Teledyne Photon Machines Inc., Bozeman, MT). The LA-system was equipped with a standard active two-volume ablation cell (HelEx II), including the

Aerosol Rapid Introduction System (ARIS, Teledyne CETAC Technologies) for fast aerosol washout. The LA unit was coupled to a quadrupole ICP-MS instrument (Agilent 7900x, Agilent Technologies, Santa Clara, CA). Ablation parameters were as follows: laser energy density, $0.84\,J\,cm^{-2}$; repetition rate, 250 Hz; beam size, 10 μm−square mask; dosage 10 and total acquisition time for ICP-MS acquisition was 0.04 s (with 12 ms dwell time for specific $^{146}Nd$ nuclide). Other parameters were based on model predictions for the fastest possible mapping times, avoidance of aliasing, minimal blur, and maximal S/N ratios[116,117]. The ablated material was transported from the ablation cell to the ICP using helium as a carrier gas and argon was added as a make-up gas before the torch of the ICP. Data processing and image analysis were performed using the software package HDIP (Teledyne Photon Machines Inc., Bozeman, MT).

## Analysis of immune cell populations in tumors using flow cytometry

Tumor tissue was harvested 14 or 15 days after the B16F10 inoculation (4 or 5 days after the first NT GSDMD treatment, respectively), cut into smaller pieces and gentleMACS Dissociator (Miltenyi Biotec) was used for the preparation of a single-cell suspension according to the manufacturer's protocol. Cells were passed through 70 μm nylon strainers (VWR) and $0.5 \times 10^6$ of tumor cells were resuspended in FACS buffer (PBS supplemented with 10% FBS). For Live/Dead staining cells were incubated on ice for 10 min in 100 μl PBS containing ZombieNIR dye (dilution 1:2000, Biolegend, 423106)). After removal of the dye, cells were resuspended in 50 μl FACS buffer containing anti-mouse CD16/CD32 (Fcγ III/II receptor) (1:25, BD Pharmingen). 10 min later, an antibody cocktail was added (50 μl) and cells were incubated on ice for at least 30 min. The antibody cocktail was prepared in FACS buffer containing True Stain Monocyte blocker (1:20, 426103, BioLegend) including the following antibodies: CD103 (clone 2E7)-BV421 (Biolegend 121421, 1:100), CD24 (clone M1/69)-APC-Fire750 (Biolegend, 101839, 1:100), CD4(clone RM4-5)-BV510 (Biolegend 100553, 1:200), CD11b (clone M1/70)-AF700 (eBioscience 56-0112, 1:200), CD8a(clone 53-6.7)-PE-Cy7 (Biolegend 100721, 1:200), F4/80(BM8)-PE-Dazzle594 (Biolegend 123145, 1:100), MHCII (cloneM5/114.15.2)-BV711 (Biolegend 107643, 1:200), CD19(clone eBio1D3)-PE (eBioscience 12-0193-82, 1:200), CD161 (NK1.1) (clone S17016D)-FITC (Biolegend 156508, 1:100), CD11c(clone N418)-BV570 (Biolegend, 117331, 1:200), CD3 (clone 145-2C11)-PE-Cy5 (Invitrogen 15-0031-82, 1:200), CD64 (clone X54-5/7.1)-PerCP-Cy5.5 (Biolegend 139307, 1:100), Ly-6C (HK1.4)-APC (Milteny 130-102-341, 1:50), Ly-6G (1A8)-BV785 (Biolegend 127645, 1:200), CD45 (30-F11)-BV605 (Biolegend 103139,1:200). As reference controls, an unstained sample and for every marker, a single-stained reference control (spleen cells) was acquired.

After incubation, cells were washed and analyzed using the Aurora spectral flow cytometer (Cytek Bioscience) with SpectroFlo v3.1.0 software (Cytek Bioscience). The resulting unmixed FCS files were analyzed using manual gating in FlowJo v. 10.10.0 software (BD Biosciences) according to the gating strategy presented in Supplementary Fig. 14. After the removal of doublets and dead cells, manual gates were imposed on different populations of CD45 positive cells, such as CD19-positive cells, CD3ε-positive cells. The CD45 + CD3ε-CD19- population was further separated into NK1.1 positive cells and Ly6G positive cells. The CD3-, CD19-, Ly6G-, and NK1.1- negative cell population was further analyzed using the UMAP dimensionality reduction (version 4.1.1)[118] method and FlowSOM (version 4.1.0)[119] clustering approach.

## Electroporation efficiency by flow cytometry

Established B16F10 tumors were electroporated with 20 μg of plasmid coding for a yellow fluorescent protein (YFP) and tumors were collected 48 h later and the single cell suspension was prepared as stated above. $2 \times 10^6$ of tumor cells were resuspended in 50 μl of FACS buffer

supplemented with 2 μl Mouse TrueSatin FcX (BioLegend) and incubated for 10 min at room temperature. Afterwards, 50 μl of antibody cocktail containing CD11c (clone N418)-BV605 (Biolegend, 117334, 1:100), CD3ε (clone 145-2C11)-BV711 (Biolegend, 100349, 1:200), CD11b (clone M1/70)-BV750 (Biolegend, 101267, 1:200), CD45 (clone 30-F11)-APC-Cy7 (Biolegend, 103115, 1:500), TRP1-Alexa Fluor 647 (abcam, ab270105, 1:100) in FACS buffer with 5 μl True Stain Monocyte blocker (BioLegend) was added and samples were incubated for at least 30 min (but no longer than 2 h) on ice. Before analysis cells were washed with FACS buffer and fluorescence was measured on an Aurora spectral flow cytometer (Cytek Biosciences). All antibodies were titrated individually according to standard practice before being used in the panel.

As reference controls, an unstained sample and, for every color, a single-stain reference control were acquired (for CD11c, CD1b, CD3, CD45 spleen cells $10^5$ and for TRP1 B16F10 cells $10^5$ were used). For YFP, B16 cells electroporated with a plasmid carrying YFP were used. All reference controls underwent the same protocol as fully stained samples, including washes.

Data were acquired and unmixed using SpectroFlo v3.1.0 software (Cytek Biosciences). The resulting unmixed FCS files were analyzed using manual gating in FlowJo (BD Biosciences) v. 10.10.0 software (BD Biosciences) according to the gating strategy in Supplementary Fig. 15. First, a manual data check was performed to ensure the exclusion of technical artifacts and bad-quality samples (clogs, doublets, and dead cells). The unmixing of raw data was performed using single-stain controls as references (no manual compensation was used). Next, doublets were removed, YFP-positive cells were gated manually, and then TRP1 and CD45-positive cells were gated.

### Statistics and reproducibility
Graphs were prepared with GraphPad Prism 8 (GraphPad Software, Boston, Massachusetts USA). All statistical analyses were performed with GraphPad Prism 8 software and are described in figure legends. One-way ANOVA was used to test statistical differences among the means of three or more populations. Two-way ANOVA with Tukey's multiple comparisons test was used for analyzing experiments with multiple groups and independent variables. For RNA-seq data Fisher's exact test was used to analyze the gene list for each sample and the mouse genome was used as the reference set. Benjamini-Hochberg test was used for multiple test adjustments. Statistical differences between survival rates of Kaplan-Meier curves were compared with the log-rank test. An unpaired two-tailed $t$-test was used to compare the two groups. Welch's correction was used in case of different variances. $P$ values of less than 0.05 were considered significant. Data with error bars are shown as mean ± SEM. Biomath (InVivoStat) and G power 3.1 (Heinrich-Heine-Universität Düsseldorf, Germany) software were used to predetermine sample size for mice experiments. Animals were randomized before treatment. No data points were excluded from the analysis. The researchers were not blinded during the experiments or data collection and analysis, except for histological, flow cytometry, and LA-ICP-MS analysis of tumor samples. It was not feasible to carry out blinding in cases when the same person was performing the experiment and analyzing the data. Blinding was performed when analyzing histological data, conducting LA-ICP-MS, and running flow cytometry of tumor samples since the person analyzing the histological slides, LA-ICP-MS data, and flow cytometry samples was not aware of the sample treatment.

### Reporting summary
Further information on research design is available in the Nature Portfolio Reporting Summary linked to this article.

### Data availability
The raw RNA sequencing data and corresponding processed data files, which underpin the findings of this study, have been deposited in the Gene Expression Omnibus under the accession code GSE261286. The remaining data supporting the findings of this study are available within the Article and Supplementary information. Source data are provided with this paper as a Source Data file. All unique biological materials are available from the corresponding author upon request.

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

## Acknowledgements

The authors thank the members of the Department of Synthetic Biology and Immunology, particularly Dr. T. Fink and Dr. E. Rihtar, for the valuable discussions. The authors thank Dr. T. Plaper for the preparation of the plasmid encoding IL-1β, E. Boršić for the preparation of plasmid encoding DmrB casp-1ΔCARD and optimizing B16F10 transfection, and K. Podgoršek and A. Perčič for the help with animal work. The authors are grateful to M. Svetlinčič, R. Bremšak, I. Škraba, K. Podgoršek, and T. Strmljan for technical assistance. We would like to thank prof. M. Čemažar (Institute of Oncology, Ljubljana, Slovenia) for 4T1 and CT26 cell lines. This research was funded by the Slovenian Research and Innovation Agency project grants J3-1746 and N3-0358 to I.H.B., Z3-4501 to T.Ž.R., N1-0377 and J7-4640 to R.J. and young researcher's PhD grant to S.O., and program funding P4-176 to R.J., and P1-0034 to M.Š. EU HORIZON-WIDERA CTGCT project 101059842 was granted to R.J. I.H.B. is grateful to the European Federation of Immunological Societies for the Eastern Star Award. I.H.B. is a member of COST PRESTO Action CA21130. S.O. is a member of COST IMMUNO-model Action CA21135.

## Author contributions

Conceptualization: S.O., I.H.B.; Methodology: S.O., T.Ž.R., Š.M., D.L., M.Š., M.B., I.H.B.; Investigation: S.O., T.Ž.R., Š.M., D.L., I.H.B.; Visualization: S.O.,. T.Ž.R., Š.M., I.H.B.; Funding acquisition: T.Ž.R., R.J., I.H.B.; Project administration: I.H.B.; Supervision: I.H.B.; Writing – original draft: S.O., I.H.B.; Writing – review & editing: S.O., T.Ž.R., Š.M., D.L., M.Š., M.B., R.J., and I.H.B.

## Competing interests

The authors declare no competing interests.
