## [Transparent Peer Review file · Nature Communications]

Cytokine-armed pyroptosis induces antitumor immunity against diverse types of tumors

Corresponding Author: Dr Iva Hafner Bratkovič

Version 0:

Reviewer comments:

Reviewer #1

(Remarks to the Author)

The authors describe pre-clinical work and cell lines in murine models the which components of pyroptosis are administered together and antitumor effects are evaluated. These include gasdermin D variants, which have pore-forming capabilities. In this study, the gasdermin variants have been modified to be constitutively active or inducible. Additional components are cytokines IL-1 , IL, 12, and IL-18. Overall, the studies are interesting and potentially important to that different components of endogenous pyroptosis may have different impacts on antitumor immune responses that may either potentiate or limit the utility of agents based on inflammasomes as anticancer therapeutics.

In one example of this, the authors note that antitumor immunity is enhanced when the gasdermin D variants are administered in combination with individual cytokines, but the same pore-forming agents combined with both IL-1 and IL-18 up regulated immunosuppressive pathways via an interferon gamma mediated mechanism abrogating antitumor effects induced by pyroptosis.

Overall, the manuscript is interesting and warrants publication in Nature Communications, although I have some general comments regarding some of the findings and potential of these data to lead to clinical advances in patients:

The authors describe regression of the primary tumor and rejection of new tumor cells on rechallenge, but there is no response in uninjected or sham-injected lesions. This is in contrast to some other injectable agents. It would be helpful to profile uninjected lesions before and after treatment of the distant site to determine whether there are any productive anti-tumor immune changes observed. If not, this would seem counterintuitive if the deconstructing inflammasome is producing actual antitumor immune responses. Why would this not create circulating antigen-experienced T cells, memory cells or SLECs? It may also be helpful to look in the peripheral blood for evidence of these. If there additional experiments are beyond the scope of the current study, it may be helpful to include some caveats or reflections in the discussion.

There is an appropriate discussion of the potential impact of optimization for future study in the discussion. It should be noted that the IL-12 exposure is almost certainly non-optimal. Repeated exposure to plasmid IL-12-EP leads to regression of both injected and non-injected lesions both clinically and preclinically. In the future, it would be helpful to replicate existing IL-12 findings and then add the GSDMD agent to the existing regimen to understand the potential for clinical synergy between IL-12 and other pyroptosis components.

The authors appropriately focus on the potential application of their techniques on discovery particularly as it pertains to optimizing the therapeutic activity of inflammasomes in combination with cytokines or checkpoint inhibitors. However, there may be significant barriers to the clinical application of the current approach. The use of plasma electroporation in patients has been attempted in patients and there have been significant barriers to the implementation and adoption of this approach. Furthermore, one of the most interesting findings was the antagonistic effect of IL-1b and IL18 with GSDMD agents. However, this antagonistic effect was abrogated by anti-PD-1 antibodies which are the mainstay of immunotherapy and clinical medicine currently. The authors may wish to allude to how a deconstructed inflammasome approach might find application in a clinical environment in which most are all eligible patients receive anti-PD-1 antibodies. It could simply be a situation which this is a platform for further discovery without clear evidence that removing aspects of the inflammasome will have direct application clinically right now.

Reviewer #2

(Remarks to the Author)

In this study, the authors developed a novel strategy named “Deconstructed Inflammasomes”, utilizing GSDMD-NT combined cytokines electroporation to induce immunogenic cell death in both immune desert and inflamed tumors, providing new tools to achieve anti-tumor immunity. Initially, the box of GSDMD comprising Full-length GSDMD and GSDMDTEV, NT GSDMD and NT GSDMD I105N were constructed to compare the ability of cell death induction. Based on the results that NT GSDMD can trigger the pyroptosis and promote tumor regression, they designed several NT GSDMD delivery models, including GSDMDTEV – split TEVp and TEVp systems. And they found NT GSDMD with cytokines, IL-1beta, IL-18, especially IL-18 can activate the antitumor immune response dependent with CD8+ and NK cells promote the survival of tumor inoculation mice. Summarily, NT GSDMD dependent deconstructed inflammasomes are thus a powerful, tunable, and tumor-agnostic strategy to enhance antitumor response to tumors.

Major points,

1. The author successfully applied the delivery system in mouse model injected with different tumor. The output provided significant destruction of tumor. Since the experiments were all performed in mice, there should be a standard to assess the efficiency of electroporation. It is essential to confirm the function and expression of GSDMD-NT after transfection. The plasmids were only imported to tumor cells or there were other potential receptor cells which determined the final killing effect especially in inflamed tumor with lymphocyte infiltration. This possibility could possibly explain much better survival rate in CT26 tumor group. Meanwhile, the data showed that cytokine combination with GSDMD-NT caused no significant related cytokine increase in sera, which maybe conflict with the fact that inflammatory response was amplified largely by means of cytokines releasing to extracellular.
2. The author's inadequate description of how to achieve the plasmid electric transfer system of cancer cells in living mice that have been previously planted with tumors is puzzling. Therefore, we cannot judge whether this system is suitable for other tumor models, because the plasmid electrotransfer system applicable to different cells is quite different, and the authors have not provided a method to detect the electrotransfer efficiency data.
3. In this study, the authors used 2 small molecules (AP21967, AP20187) as the activator of the deconstructed inflammasomes system, and the effect looks good. However, as potential tumor therapy strategies, the biosafety of these small molecules has not been verified by relevant physiological toxicity experimental data.
4. In many mice model experimental scheme of this study plasmid electroporation were performed followed by the implantation of tumor cells (Day 0, Day 1), I wondered whether the process of electroporation of different plasmids itself would affect the normal development of tumor cells. In particular, in practical applications, most cancers are detected when the tumor is large, so will the efficiency of treatment using this system be significantly affected?
5. In figure 4b-d, CD8+ or NK deletion Ab were treated before deconstructed inflammasomes electroporation, I found then the mice could survive for at least 10 days, even 20 days. While in experiments of other figures, the mice all began to die at less than 5 days. Why did this happen?

Minor points:

1. In supplemented 1f, increasing amounts of plasmids transfected led to more cell death even without AP treatment. The authors need to provide data showing the expression upon transfection.
2. In mouse tumor model, the survival% and tumor size were analyzed while the weight loss% was not shown?
2. B16F10 and 4T1 were selected to inoculated into mouse in this study. Why the authors chose these two types of cells instead of other tumor cells?
3. As for antitumor immune response, the CD8+T cells and NK cells were analyzed. How about other infiltrating immune cells?
4. When mouse was inoculated with NT-GSDMD with cytokines, the tumor size was reduced and tumor regression was inhibition. How about host systemic inflammation level?
5. The P value or differences need to be marked.

Reviewer #3

(Remarks to the Author)

The manuscript of Orehek and colleagues describe synthetic approaches to trigger pyroptosis in vivo in engrafted mouse cancer models in order to re-activate the anti-tumoral immunity. They first report different technical approaches to induce pyroptosis in tumor cells, then focus on the impact of electrogenic gene transfer of the active fragment of GSDMD in inducing pyroptosis in vitro and in vivo. They further test the impact of adding inflammasome dependent cytokines production to the anti-tumoral properties of pyroptosis. Although the topic and approaches are important to the field of cancer research, the described work lacks many controls, overstate the results and for some parts the authors have troubles to organize the information in a clear way for the reader. I recommend to reject the manuscript.

First, the term used throughout the manuscript of deconstructed inflammasome is incorrect since the authors manipulate the inflammasome outputs and not the proteins composing the inflammasome. The title and text must be corrected.

Second, it is not clear what is the interest of presenting all the different technical approaches they used nor the development of the DmrB caspase-1.

Third the manuscript lacks many important controls and many results do not support the claims by the authors. The tumor

growth curves in figure 1e show that most of the mice injected do not have a delayed tumor growth. Only 3 animals controlled tumor growth and the statistical analysing is not supporting a significant effect , P=0.07. Data on figure 2g are also not statistically significant but at least the individual growth curve highlight a delay in tumor growth in many animals. Why the authors selected NT-GSDMD to perform the experiments of their manuscript?

There is no control demonstrating that the in vivo anti-tumoral effect is via the action of GSDMD within tumor cells and not due to incorporation in other cell types.

Figure 3c-d lack the group of mice showing the response to GSDMD treatment alone, without this data the authors cannot conclude that adding any cytokine to the mix will modulate the efficiency of pyroptosis.

Figure 4, there is no statistical significance between the growth curves

Figure 5 It is not clear why the authors did not compare the tumor growth curve of NT GSDMD+IL-1+IL-18 to the tumor growth curve of NT GSDMD+IL1 alone and NT GSDMD +IL-18 alone, but to IL-12.

Figure 6 many control groups are missing to be able to compare the changes in the response: panel e: the NT-GSDMD is missing

Panel h: the NT-GSDMD+PD1, the NT-GSDMD+IL-1+PD1, NT-GSDMD+IL-18+PD1. In fact the response of NT-GSDMD+IL-1+IL-18+PD1 looks similar to NT-GSDMD+IL-1+IL-18 alone. Moreover the legend must be NT-GSDMD+ IL-1+IL-18 and not IL-1+IL-18.

Minor points.

Concerning the in vivo gene transfer the material and method description does not allow to reproduce the technique. How the electroporation is performed into tumors?

Lane116: the gasdermin facilitates pyroptosis not necrosis.

Lane 446 clarify "regardless of the GSDMD system used"

Reviewer #4

(Remarks to the Author)

The manuscript by Orehek et al. described a very interesting methods for the synthetic induction of pyroptosis plus inflammatory cytokines for the induction of immune-mediated cancer treatment. To support this conclusion, the authors should include a setting with a proper pyroptosis inhibitor, e.g. disulfiram to prove that all effects observed are indeed mediated by pyroptosis induction. In addition, the authors designed this complicated way for inducing cancer cell pyroptosis by i.t. injection of plasmids and in-situ electroporation, this should be compared with direct i.t. injection of pyroptosis inducers, based on which the authors should extend the translational value of such synthetic induction. Some issues about experimental design is detailed below:

- 1) Figure1&S1: The authors should provide data about the efficiency of i.t. plasmid electroporation. In addition, to confirm this tumor-regressing effect is dependent on the immune system, the same settings should be applied on nude mice or on C57Bl/6 with CD8 depletion. Without this, the conclusion described in Text Line 163-164 could not yet be supported.
- 2) Figure 2, the same suggestion, the experiment should be performed in T cell deficient mice to confirm the involvement of antitumor immunity. Additional controls should be considered, such as electroporation of empty plasmid and i.t. injection of solvent. The number of animals in each group should be increased to ensure sufficient statistical analysis.
- 3) Figure 3, the effect of electroporated IL1b plasmid should be compared with i.t. injection of recombinant IL1b so as to better discuss the conclusion and the debate in line 235-236.
- 4) Many in vivo experimental comparisons were performed cross experiments, for example the data obtained in Figure 2 and 3 were compared directly with Figure 1 (line 200, line 229-230), instead of appropriate replication of the observations in Figure 1, this is unacceptable, especially considering the relative low mice numbers in each group. An example of this issue could be a direct inconsistency between figure 4c (where the NT-GSDMD + Isotype IgG obtained ~50% survival) and figure 1 (NT-GSDMD obtained ~25% survival)! This indicates the importance to have all indispensable controls in all experiments.
- 5) Figure 4, A, necessary labelling should be added to annotate different immune cells. Experiments in C& D miss all necessary controls (all groups without electroporation)
- 6) The isotype control for anti-PD1 and anti-CTLA4 is not correct. Clone RMP1-14 is Rat IgG2a, Clone 9D9 is a Mouse IgG2b, but the authors selected a mouse IgG2a isotype control, this should be addressed.
- 7) The authors described an in vivo bioluminescence method (Line 645), and bioluminescence images in the supplement figures, but it seems that is no such data provided in the main figures and text. Were all tumor sizes quantified by luminescence or volume (as depicted in all tumor growth curves in the main figures)?

Version 1:

Reviewer comments:

Reviewer #1

(Remarks to the Author)

My concerns were adequately addressed.

Reviewer #2

(Remarks to the Author)

The authors have addressed my concerns. I have no further questions and support the publication.

Reviewer #3

(Remarks to the Author)

The manuscript has been greatly improved and is now suitable for publication.

Reviewer #4

(Remarks to the Author)

The authors have addressed all my concerns with new experiments and improved discussions. I have no more suggestions.

Reviewer #1 (Remarks to the Author): with expertise in melanoma, cancer immunology, plasmid electroporation

The authors describe pre-clinical work and cell lines in murine models the which components of pyroptosis are administered together and antitumor effects are evaluated. These include gasdermin D variants, which have pore-forming capabilities. In this study, the gasdermin variants have been modified to be constitutively active or inducible. Additional components are cytokines IL-1 β , IL, 12, and IL-18. Overall, the studies are interesting and potentially important to that different components of endogenous pyroptosis may have different impacts on antitumor immune responses that may either potentiate or limit the utility of agents based on inflammasomes as anticancer therapeutics.

In one example of this, the authors note that antitumor immunity is enhanced when the gasdermin D variants are administered in combination with individual cytokines, but the same pore-forming agents combined with both IL-1 β and IL-18 up regulated immunosuppressive pathways via an interferon gamma mediated mechanism abrogating antitumor effects induced by pyroptosis.

Overall, the manuscript is interesting and warrants publication in Nature Communications, although I have some general comments regarding some of the findings and potential of these data to lead to clinical advances in patients:

R1.1 The authors describe regression of the primary tumor and rejection of new tumor cells on rechallenge, but there is no response in uninjected or sham-injected lesions. This is in contrast to some other injectable agents. It would be helpful to profile uninjected lesions before and after treatment of the distant site to determine whether there are any productive anti-tumor immune changes observed. If not, this would seem counterintuitive if the deconstructing inflammasome is producing actual antitumor immune responses. Why would this not create circulating antigen-experienced T cells, memory cells or SLECs? It may also be helpful to look in the peripheral blood for evidence of these. If there additional experiments are beyond the scope of the current study, it may be helpful to include some caveats or reflections in the discussion.

We are grateful for the reviewer's appreciation of the study and the comments that helped us to clarify the immunological aspects of the treatment and improve the discussion on the translational potential of the developed concept.

A1.1: Regarding the effectivity of our and other similar systems where the induction of cell death did not lead to complete rejection of distant tumor formation (e.g. for B16 melanoma Van Hoecke, 2018, PMID 30143632; for EMT6 tumors Fontana, 2024, PMID 9243763), one reason for apparently lower functionality of those studies could be in the different experimental setup. We rechallenged only mice that responded to the treatment, whereas the mentioned studies injected tumors prior mice were cured, thus also mice where treatment was not effective were included. We modified the scheme of the rechallenge experiment (New Fig. 9a) to clarify that only treatment-responding animals or naïve animals were injected in the second flank. As the reviewer suggested in the revised version of the manuscript we reflect on some limitations of the study, but moreover we now provide several levels of evidence that the functional

immune system is crucial for the antitumor effects of cytokine-armed pyroptosis. In addition to already reported immune cell population depletion (new Supplementary Fig. 11) and rechallenge experiment (new Fig. 9a - e), we now also demonstrate that the treatment fails to retard tumor growth in immunocompromised mice (new Fig. 8a - c). We used ultra immunodeficient B-NDG knockout mice that lack B cells, T cells, and also functional NK cells. To deepen our understanding of possible immune cells involved, we profiled immune cells in mock-treated (empty vector) and armed pyroptosis-treated tumors. Particularly, we observed an increase in T-cell and NK1.1+ cell populations in the case of armed pyroptosis treatment (Fig. 8e, Supplementary Fig. 11c). All these experiments demonstrate the involvement of immune system, particularly T and NK cells in mediating the protective effects of cytokine-armed pyroptosis on primary tumors or systemically (rechallenge experiment) and are in line with other studies reporting on stimulation of anti-tumor immunity by cancer cell death (Zhang, 2020, PMID: 32188940; Wang, 2020, PMID 32188939, Fontana, 2024, PMID 9243763).

R1.2 There is an appropriate discussion of the potential impact of optimization for future study in the discussion. It should be noted that the IL-12 exposure is almost certainly non-optimal. Repeated exposure to plasmid IL-12-EP leads to regression of both injected and non-injected lesions both clinically and preclinically. In the future, it would be helpful to replicate existing IL-12 findings and then add the GSDMD agent to the existing regimen to understand the potential for clinical synergy between IL-12 and other pyroptosis components.

A1.2: For the purpose of our study, we only used a single IL-12 electroporation to investigate if such treatment without the addition of NT GSDMD is potent enough to induce tumor regression and antitumor effect. The inefficiency of single-electroporation of IL-12 encoding DNA without other interventions corroborates previous preclinical B16 melanoma studies (e.g. Lucas, 2002, PMID 12027550; Kamenšek, 2018, PMID 29468364) that already demonstrated that more than one electroporation of IL-12 plasmid is needed to achieve the protective effect. Furthermore, the protocols in clinical studies use two or more rounds of IL-12 electroporation treatments (NCT01579318, NCT00323206, NCT02345330, NCT01502293). We agree that in the future it will be interesting to see how pyroptosis induction is complementing the existing IL-12 electroporation regimens. To emphasize the difference between our and previous IL-12 electroporation treatments we rewrote the corresponding paragraphs in the results and discussion sections.

R1.3 The authors appropriately focus on the potential application of their techniques on discovery particularly as it pertains to optimizing the therapeutic activity of inflammasomes in combination with cytokines or checkpoint inhibitors. However, there may be significant barriers to the clinical application of the current approach. The use of plasma electroporation in patients has been attempted in patients and there have been significant barriers to the implementation and adoption of this approach.

Furthermore, one of the most interesting findings was the antagonistic effect of IL-1b and IL18 with GSDMD agents. However, this antagonistic effect was abrogated by anti-PD-1 antibodies which are the mainstay of immunotherapy and clinical medicine currently.

The authors may wish to allude to how a deconstructed inflammasome approach might find application in a clinical environment in which most are all eligible patients receive anti-PD-1 antibodies. It could simply be a situation which this is a platform for further discovery without clear evidence that removing aspects of the inflammasome will have direct application clinically right now.

A1.3. Taking the reviewer's suggestions into account we refined the discussion to include several advancements in plasmid electroporation in cancer therapy as well as studies that use the modulation of inflammasomes as antitumor therapy and discuss the potential further development of cytokine-armed pyroptosis as a combination with ICI therapy. As an independent study we are currently investigating the effect of different cell death inducers in combination with anti-PD1 therapy and the effects of NT GSDMD introduction with anti-PD1 therapy are very exciting, with the majority of mice going into remission after B16 melanoma induction.

Reviewer #2 (Remarks to the Author): with expertise in pyroptosis/gasdermins, immunology

In this study, the authors developed a novel strategy named "Deconstructed Inflammasomes", utilizing GSDMD-NT combined cytokines electroporation to induce immunogenic cell death in both immune desert and inflamed tumors, providing new tools to achieve anti-tumor immunity. Initially, the box of GSDMD comprising Full-length GSDMD and GSDMDTEV, NT GSDMD and NT GSDMD I105N were constructed to compare the ability of cell death induction. Based on the results that NT GSDMD can trigger the pyroptosis and promote tumor regression, they designed several NT GSDMD delivery models, including GSDMDTEV – split TEVp and TEVp systems. And they found NT GSDMD with cytokines, IL-1 β , IL-18, especially IL-18 can activate the antitumor immune response dependent with CD8 $^+$ and NK cells promote the survival of tumor inoculation mice. Summarily, NT GSDMD dependent deconstructed inflammasomes are thus a powerful, tunable, and tumor-agnostic strategy to enhance antitumor response to tumors.

Major points,

R2.1 The author successfully applied the delivery system in mouse model injected with different tumor. The output provided significant destruction of tumor. Since the experiments were all performed in mice, there should be a standard to assess the efficiency of electroporation.

We would like to thank the reviewer for constructive comments that helped us to improve the manuscript.

A2.1 In order to measure electroporation efficiency, B16F10 tumors were electroporated with the plasmid encoding the yellow fluorescent protein. Afterward, flow cytometry was performed to determine the % of electroporated cells. We show that in our experimental setup, the electroporation efficiency was approximately 3.4% (Supplementary Fig. 3b).

R2.2 It is essential to confirm the function and expression of GSDMD-NT after transfection.

A2.2 We provide additional experiments confirming the expression and function of NT GSDMD upon transfection of B16F10 cells: the expression of NT GSDMD was followed by western blot (Fig. 2c), induction of pyroptosis was measured by following PI uptake (Fig. 2b). Using confocal microscopy, we also demonstrated that transfection of NT GSDMD caused pyroptosis, characterized by ballooning-like morphology (Fig. 2a and Supplementary Fig. 2). Upon transfection of B16F10 cells with the inducible dDmrB casp-1:GSDMD single-component system, we demonstrate that addition of the dimerizer AP20187, which induces activation of caspase-1 that causes cleavage and thereby activation of gasdermin D and cell death (Supplementary Fig. 5d, e).

R2.3 The plasmids were only imported to tumor cells or there were other potential receptor cells which determined the final killing effect especially in inflamed tumor with lymphocyte infiltration. This possibility could possibly explain much better survival rate in CT26 tumor group.

A2.3 Using reporter electroporation of B16F10 melanoma tumors, followed by flow cytometry, we show that the major electroporated cell type (95%) are B16F10 melanoma cells (Supplementary Fig. 3c). Supplementary to the depletion experiment we provide additional experiments demonstrating that functional lymphocytes are crucial for the therapeutic effect as immunocompromised mice that lack B, T and functional NK cells do not respond to the treatment (Fig. 8a - c). Electroporation of tumor cells followed by their cell death stimulates immune cells that induce tumor eradication. As commented by the reviewer, in the inflamed tumor, a number of lymphocytes are present that could readily respond to tumor cell death, thus likely the effect of the treatment is so robust.

R2.4 Meanwhile, the data showed that cytokine combination with GSDMD-NT caused no significant related cytokine increase in sera, which maybe conflict with the fact that inflammatory response was amplified largely by means of cytokines releasing to extracellular.

A2.4 We first demonstrated that the plasmids upon transfection in vitro induced the release of particular cytokine (Fig. 4a for IL-1 β and IL-18; and Fig. 5a for IL-12). Although in vivo the systemic levels of cytokines were not increased, the biological effect of cytokine-armed pyroptosis was still obvious. As we reflect in the discussion, local delivery of cytokine is beneficial, as particularly high systemic levels of IL-12 were previously connected with adverse side effects and the major problems of IL-12 therapy.

R2.5 The author's inadequate description of how to achieve the plasmid electric transfer system of cancer cells in living mice that have been previously planted with tumors is puzzling. Therefore, we cannot judge whether this system is suitable for other tumor models, because the plasmid electrotransfer system applicable to different cells is quite different, and the authors have not provided a method to detect the electrotransfer efficiency data.

A2.5: We provide a detailed description of electroporation in the methods section including schematic representation (Supplementary Fig. 3a). As mentioned earlier we now also provide a method for the determination of electroporation efficiency using fluorescent protein as a marker (Supplementary Fig. 3b). We should note however, that different electroporation systems have been developed and are used in the clinic, including systems with multiple electrode pins that are inserted intratumorally to provide better electroporation efficiency, in addition to other delivery systems that could be used instead of electroporation such as lipid nanoparticles. We mention some of those systems in the discussion section.

R2.6: In this study, the authors used 2 small molecules (AP21967, AP20187) as the activator of the deconstructed inflammasomes system, and the effect looks good. However, as potential tumor therapy strategies, the biosafety of these small molecules has not been verified by relevant physiological toxicity experimental data.

A2.6: We used AP21967 to demonstrate the functionality of a heterodimeric system in vitro. In vivo, we used homodimerizer AP20187, the compound also known as B/B homodimerizer that has been used in mouse models for more than 20 years (e.g. some early references Xie, 2001, PMID: 11559553, Hanks, 2005, PMID:15665830, Burnett, 2004, PMID:14726498). We did not observe increased weight loss or abnormal blood analysis values in mice receiving AP20187 compared to others (Supplementary Fig. 5 g,

h). Another reason for choosing AP20187 is that this compound does not bind with endogenous FKBP and thus does not have immunosuppressive activity (through e.g. mTORC1 inhibition (Chang, 2003, PMID: 12554853, Yang, 2000, PMID: 10737745)).

R2.7: In many mice model experimental scheme of this study plasmid electroporation were performed followed by the implantation of tumor cells (Day 0, Day 1), I wondered whether the process of electroporation of different plasmids itself would affect the normal development of tumor cells. In particular, in practical applications, most cancers are detected when the tumor is large, so will the efficiency of treatment using this system be significantly affected?

A2.7: In all our experiments, we first seed the tumor cells, and after 8-10 days, when tumors are relatively large (longest dimension cca 5-9 mm), we perform electroporation treatment. So this approach is analogous to late treatments of relatively large tumors. We believe that treating smaller tumors or using a more effective delivery system should improve the effect, particularly for the delivery of cytokine DNA, while for the delivery of cell death inducer, this system is potent enough, as only a minor fraction of pyroptotic cells is enough for tumor elimination mediated by the immune system (Wang, 2020, PMID 32188939, Fontana, 2024, PMID 9243763). We discuss some limitations as well as possible improvements to our study in the discussion section.

R2.8

In figure 4b-d, CD8+ or NK deletion Ab were treated before deconstructed inflammasomes electroporation, I found then the mice could survive for at least 10 days, even 20 days. While in experiments of other figures, the mice all began to die at less than 5 days. Why dis this happen?

A2.8: In the depletion experiment, mice had to be sacrificed even earlier as tumors reached the defined humane endpoint. In the first submission, day 0 was defined a bit differently among different experiments. To avoid confusion, we modified survival curves in the figures in a way that day 0 is the day of the first electroporation, similar to when patients are followed in clinical trials.

Minor points:

R2.9 In supplemented 1f, increasing amounts of plasmids transfected led to more cell death even without AP treatment. The authors need to provide data showing the expression upon transfection.

A2.9: To develop a bicistronic single-component system that is inducible upon the addition of a small molecule, we designed two constructs that differ in the orientation of DmrB casp-1 and GSDMD. As noted by the reviewer, the construct with GSDMD preceding DmrB casp-1 indeed has substantial activity even in the absence of the AP20187 ligand (Supplementary Fig. 5c). We now provide a western blot demonstrating that cleaved NT GSDMD is also present in the absence of AP20187 (Supplementary Fig. 5e). We are however stressing that this construct has not been used further in animal experiments.

R2.10 In mouse tumor model, the survival% and tumor size were analyzed while the weight loss% was not show?

A2.10 Weight loss was followed in all experiments as it is an important indicator of animal well-being (the defined humane endpoint is 20% weight loss). We should note that none of the animals were sacrificed based on this criterion. We now provide animal body weights (normalized to the body weight on day of

the first electroporation) followed during the experiment in Supplementary Figures 3d, 5g, 6a, 6c, 11a, 11b, 12c, 13d.

R2.11 B16F10 and 4T1 were selected to inoculated into mouse in this study. Why the authors chose these two types of cells instead of other tumor cells?

A2.11. We were particularly interested in whether the developed approach that does not depend on the expression of inflammasome components can be used to treat the most resilient types of tumors e.g. immune-excluded (cold) tumors that do not respond to immune checkpoint blockade. B16F10 melanoma as well as 4T1 breast cancer models are mouse models of cold tumors.

R2.12 As for antitumor immune response, the CD8+T cells and NK cells were analyzed. How about other infiltrating immune cells?

A2.12 In the revised manuscript we include two new experiments that provide insight into the involvement of the immune system in tumor clearance. Using ultra-immunodeficient B-NDG knockout mice that lack B, T, and NK cells we demonstrate that armed pyroptosis does not affect tumor growth (Fig. 8a - c) in the absence of immune cells. Furthermore, as suggested by the reviewer we also analyzed CD45+ populations in tumors after treatment with NT GSDMD and IL-1 β compared to empty vector-electroporated tumors (Fig. 8e, Supplementary Fig. 11c). We show that particularly NK1.1+ cells (NK) and T cells were upregulated in the armed pyroptosis treatment.

R2.13 4. When mouse was inoculated with NT-GSDMD with cytokines, the tumor size was reduced and tumor regression was inhibition. How about host systemic inflammation level?

We measured respected cytokine values in the blood and did not observe an increase compared to other treatments (Supplementary Fig. 6e). Additionally, we did not observe differences in the levels of IFN γ or IL-6, the markers of cytokine release syndrome (Supplementary Fig. 6f). This suggests that single electroporation of plasmids encoding cytokines does not lead to an increase in systemic cytokine concentrations and systemic inflammation.

R2.14 5. The P value or differences need to be marked.

Statistical analysis was performed for the majority of experiments and described in the methods section. P values are depicted in the figures.

Reviewer #3 (Remarks to the Author): with expertise in inflammasome, pyroptosis/gasdermins, cancer

The manuscript of Orehek and colleagues describe synthetic approaches to trigger pyroptosis in vivo in engrafted mouse cancer models in order to re-activate the anti-tumoral immunity. They first report different technical approaches to induce pyroptosis in tumor cells, then focus on the impact of electrogenic gene transfer of the active fragment of GSDMD in inducing pyroptosis in vitro and in vivo. They further test the impact of adding inflammasome dependent cytokines production to the anti-tumoral properties of pyroptosis. Although the topic and approaches are important to the field of cancer research, the described work lacks many controls, overstate the results and for some parts the authors have troubles to organize the information in a clear way for the reader. I recommend to reject the manuscript. We thank the reviewer for his/her comments and for acknowledging the importance of the approach for

cancer research. We included additional experiments and rewrote the manuscript to improve the flow and presentation/interpretation of results.

R3.1 First, the term used throughout the manuscript of deconstructed inflammasome is incorrect since the authors manipulate the inflammasome outputs and not the proteins composing the inflammasome. The title and text must be corrected.

A3.1 Deconstruction (not reconstruction) is an analytic technique used in philosophy and literary science. In modern cuisine, the crucial components of a traditional dish are prepared separately and plated together as a deconstructed dish that retains the concept and flavors of the original dish. Deconstructed inflammasomes although as pointed out by the reviewer do not consist of inflammasomes, however, expose both inflammasome activation arms, cell death, and cytokine release.

We thank the reviewer for pointing this out and agree that the term might bring confusion thus we followed the reviewer's suggestion, we changed the title, and used cytokine-armed pyroptosis as the general term describing the approach. We retained the term deconstructed inflammasome when cytokines related to inflammasome were used in combination with gasdermin D.

R3.2 Second, it is not clear what is the interest of presenting all the different technical approaches they used nor the development of the DmrB caspase-1.

A3.2: We designed several different ways how to induce pyroptosis using GSDMD, where pyroptosis is regulated only by transcription (NT GSDMD) or posttranslationally, by cleavage with proteases that can be either their native activators (caspase-1) or exogenous (TEVp). In addition to the hypomorph GSDMD variant that kinetically delays cell death, we also prepared a single-component system, where pyroptosis is induced by the addition of a small molecule. We will make this toolbox of pyroptosis effectors available to the community, and we believe it is a valuable tool for further exploration that can, when integrated into synthetic biology networks and logic operations, enable the specific killing of cancer cells. In the revised version of the manuscript we also profiled cancer cell lines for the expression of NLRP3 inflammasome proteins and demonstrated that canonical NLRP3 activators could not function in any of those cell lines, further strengthening the rationale for the preparation of the synthetic toolbox.

R3.3 Third the manuscript lacks many important controls and many results do not support the claims by the authors. The tumor growth curves in figure 1e show that most of the mice injected do not have a delayed tumor growth. Only 3 animals controlled tumor growth and the statistical analysing is not supporting a significant effect , $P=0.07$.

A3.3 As would be expected for late-time human patient treatment, we are treating relatively large tumors (5 - 9 mm) of variable volumes. Our experiments demonstrate that even some large-size tumors respond to such treatment, while the rest of the animals needed to be sacrificed as the humane endpoint was reached (12 mm). We performed additional experiments including all relevant groups and now demonstrate a statistically significant effect of electroporation of plasmid encoding GSDMD compared to empty vector-electroporation.

R3.4 Data on figure 2g are also not statistically significant but at least the individual growth curve highlight a delay in tumor growth in many animals. Why the authors selected NT-GSDMD to perform the experiments of their manuscript?

A3.4 In addition to previously shown long-term remission of 25% of mice (Supplementary Fig. 5f), we with additional experiments now demonstrate that inducing pyroptosis with small molecule AP20187 significantly enhances the survival of B16 melanoma-bearing mice compared to electroporated mice without AP20187 or mock-electroporated mice treated with AP20187 (Fig. 3e - g).

Gasdermins are very powerful cell death inducers. In fact, according to one of the discoverers of the gasdermin family, Feng Shao, cell death can be detected earlier than NT GSDMD fragment can be visualized using western blot. Among gasdermins, Gasdermin D was chosen due to its connection to inflammasomes and our expertise in the regulation of GSDMD pore formation (PMID: 34289345, PMID: 36662620). Other gasdermins with introduced casp-1 or TEV protease cleavage sites could be tested in a similar setup. We explain the rationale for choosing gasdermin D a bit more in the results section.

R3.5 There is no control demonstrating that the in vivo anti-tumoral effect is via the action of GSDMD within tumor cells and not due to incorporation in other cell types.

A3.5 We now include an experiment demonstrating that electroporation (using a plasmid encoding yellow fluorescent protein) primarily targets B16F10 cells. Among electroporated cells, 95% are positive for TRP-1, a marker of melanoma, and less than 4% are CD45+ positive (Supplementary Fig. 3b, c). Other studies (included in the discussion) showed that inflammasome activation in noncancer cells might actually promote tumor progression, thus the targeting tropism when incorporating cell death inducers is an important aspect of our approach.

R3.6 Figure 3c-d lack the group of mice showing the response to GSDMD treatment alone, without this data the authors cannot conclude that adding any cytokine to the mix will modulate the efficiency of pyroptosis.

A3.6 Taking into account reviewer's comments regarding the flow of the study the initial submission's figures 3 and 5 have been significantly rearranged. We performed a novel experiment with a side-by-side comparison of NT GSDMD and NT GSDMD+IL-12 treatment and demonstrated that the addition of this cytokine significantly increases the median survival of mice compared to pyroptosis alone (Figure 5).

R3.7 Figure 4, there is no statistical significance between the growth curves

A3.7 Upon rearranging the manuscript, this comment now addresses the new Supplementary Fig. 11d - f. The comment is addressing the experiment where we depleted CD8+ and NK1.1+ cells. In addition to this experiment, we analyzed CD45+ cell populations in NT GSDMD+IL1 β and empty vector-treated tumors, and show increased levels of CD3+, particularly CD8+ population as well as in NK1.1+ population (Fig. 8e and Supplementary Fig. 11c). Moreover, in the revised version, we demonstrate that cytokine-armed pyroptosis approach does not work in ultra-immunodeficient B-NDG knockout mice that are deficient in B, T, and NK cells (Fig. 8a - c). Using several complementary approaches (in addition to the rechallenge experiment in Fig. 9a - e) we prove that a functional immune system employing NK and T cells is crucial for the protective effect of cytokine-armed pyroptosis.

R3.8 Figure 5 It is not clear why the authors did not compare the tumor growth curve of NT GSDMD+IL-1+IL-18 to the tumor growth curve of NT GSDMD+IL1 alone and NT GSDMD +IL-18 alone, but to IL-12.

A3.8: Although all these experiments were performed at the same time, we did not include the combination in previous Fig.4 not to overcrowd it. We agree with the reviewer and have rewritten this part. We now include all inflammasome-related treatments in Fig. 4 and appropriately compare the combination of cytokines to single-cytokine pyroptosis treatment. We introduce IL-12 in a separate Fig. 5.

R3.9 Figure 6 many control groups are missing to be able to compare the changes in the response:

panel e: the NT-GSDMD is missing

A3.9 We performed a new experiment including appropriate control, demonstrating that combining IFN γ with NT GSDMD abolishes the effect of pyroptosis alone (Fig. 7c).

R3.10

Panel h: the NT-GSDMD+PD1, the NT-GSDMD+IL-1+PD1, NT-GSDMD+IL-18+PD1. In fact the response of NT-GSDMD+IL-1+IL-18+PD1 looks similar to NT-GSDMD+IL-1+IL-18 alone. Moreover the legend must be NT-GSDMD+ IL-1+IL-18 and not IL-1+IL-18.

A3.10: We thank the reviewer for noticing the error in the legend (which we have corrected). These results are now presented in Fig. 7e - g. The rationale of this experiment is to check if immune checkpoints mediate the inhibitory effect of NT GSDMD+IL-1 β +IL-18, where all animals need to be rapidly sacrificed. In terms of long-term survival, 50% of NT GSDMD+IL-1 β +IL-18+antiPD1-treated mice survived, while only anti-PD1 therapy with empty vector-treated mice had no effect (0% survival). While we include the isotype controls for anti-PD1 and anti-CTLA4 antibodies, we believe that the suggested experiments are out of the scope of the current study, but a great idea for further development of the approach. We can also mention that we are currently conducting a parallel study addressing the effects of different cell death inducers in combination with anti-PD1 therapy and in the case of anti-PD1 + NT GSDMD electroporation the majority of B16 melanoma mice went into remission.

Minor points.

R3.11 Concerning the in vivo gene transfer the material and method description does not allow to reproduce the technique. How the electroporation is performed into tumors?

We apologize for not providing enough information. We now introduce a detailed protocol for the introduction of plasmid DNA into tumors via electroporation, including a scheme in the supplement (Supplementary Fig. 3a).

R3.12 Lane116: the gasdermin facilitates pyroptosis not necrosis.

We corrected this.

R3.13 Lane 446 clarify "regardless of the GSDMD system used".

We rewrote this sentence to make it more clear.

Reviewer #4 (Remarks to the Author): with expertise in cancer immunology, immunogenic cell death

R4.1 The manuscript by Orehek et al. described a very interesting methods for the synthetic induction of pyroptosis plus inflammatory cytokines for the induction of immune-mediated cancer treatment. To support this conclusion, the authors should include a setting with a proper pyroptosis inhibitor, e.g. disulfiram to prove that all effects observed are indeed mediated by pyroptosis induction.

A4.1 We thank the reviewer for the suggestions that helped us to improve the quality of the study. The reviewer suggests the use of a pyroptosis inhibitor to validate the findings, which is a valuable suggestion. However, gasdermin inhibitors, particularly compounds used for the treatment of various diseases such as disulfiram and dimethyl fumarate are cysteine-reacting compounds and have a potent antitumor activity that is independent of gasdermin D (e.g. Fontes, 2022, PMID: 36358950 demonstrates the antitumor activity of disulfiram in B16 melanoma model, disulfiram entered clinical trials for various types of cancer including metastatic melanoma and DMF for blood cancers and glioblastoma multiforme). The use of gasdermin inhibitors might thus lead to tumor regression due to their pleiotropic activity. In the revised version, we however add several levels of proof for pyroptosis induction with our constructs. We first demonstrate that the introduction of NT GSDMD into B16F10 tumor cells in vitro leads to pyroptosis with microscopy, LDH, and PI uptake (Fig. 2a - c). We further demonstrate an increased PI uptake in tumors electroporated with NT GSDMD compared to the control (Fig. 2g and Supplementary Fig. 4). We also demonstrate the functionality of the inducible system in B16F10 cells, where the addition of the dimerizer induced production of NT GSDMD and cell death (Supplementary Fig. 5d, e). Probably the best proof that pyroptosis induction is responsible for the observed effect is the chemically inducible system, where in the absence of dimerizer AP20187, the system is not inducing tumor regression. We added the additional control (empty vector electroporation + AP20187) to demonstrate that AP20187 in the conditions used does not cause tumor regression.

R4.2 In addition, the authors designed this complicated way for inducing cancer cell pyroptosis by i.t. injection of plasmids and in-situ electroporation, this should be compared with direct i.t. injection of pyroptosis inducers, based on which the authors should extend the translational value of such synthetic induction.

A4.2: Pyroptosis is induced upon activation of inflammasomes that were shown to have tumor-suppressing or tumor-promoting actions. The effect of inflammasome activators as antitumor agents will thus depend on the expression of inflammasome components (inflammasome sensor e.g. NLRP3, adaptor ASC, and the full-length caspase-1) and gasdermin D. We aim to provide tumor-agnostic therapy that does not depend on the expression profile of inflammasome components. In the initial submission, we profiled tumor cell lines for the expression of gasdermin D and we showed it is not expressed in B16F10 cells. Now we also profiled those cell lines for expression of NLRP3, ASC, and caspase-1 and show that NLRP3 inflammasome or other ASC-dependent inflammasomes could not be activated in any of those tumor cells as either caspase-1 was not expressed (B16F10) or ASC was not expressed (4T1, CT26) (Supplementary Fig. 1). When used, inflammasome activators will likely target other cell types, particularly dendritic cells. AIM2 inflammasome in dendritic cells promotes regulatory T-cell recruitment and melanoma progression (Fukuda, 2021, PMID: 34325468). Dendritic cell pyroptosis might abort tumor-protective effects (Zhivaki,

2020, PMID: 33207188). We further address this point in the discussion. Experimentally, we additionally profiled which types of cells are electroporated and demonstrated that the vast majority (95%) of electroporated cells belong to cancer cells (Supplementary Fig. 3c). In September, when the revision was in the final stage of preparation, Wu and Lieberman published a study, describing a direct GSDMD activator, that potently suppressed tumor growth, but only if cancer cells expressed GSDMD, corroborating our conclusions (Fontana, 2024, PMID: 39243763).

Some issues about experimental design is detailed below:

R4.3: Figure 1&S1: The authors should provide data about the efficiency of i.t. plasmid electroporation. In addition, to confirm this tumor-regressing effect is dependent on the immune system, the same settings should be applied on nude mice or on C57Bl/6 with CD8 depletion. Without this, the conclusion described in Text Line 163-164 could not yet be supported.

A4.3: We used the electroporation of plasmid encoding the yellow fluorescent protein to determine the electroporation efficiency of large tumors. We show that using our approach we electroporate less than 5% (3.4%) of cells in the tumor (Supplementary Fig. 3b), however, the large majority (95%) of electroporated cells are positive for the melanoma marker (Supplementary Fig. 3c). Wang and coworkers (Wang, 2020, PMID: 32188939) previously demonstrated that only a small fraction of tumor cells need to die at the first stage for complete eradication of tumor, which is supported by our study. In addition to the depletion and rechallenge experiments, we corroborate the crucial role of the immune system in mediating the protective effects of armed pyroptosis with a novel experiment suggested by the reviewer using ultra-immunodeficient B-NDG knockout mice that are deficient in B, T, and NK cells, where the treatment is ineffective (Fig. 8a - c). Additionally, we analyzed CD45⁺ cell populations where we observed an increase in CD3⁺ cells, particularly CD8⁺ cells and also NK1.1⁺ cells when tumors were treated with NT GSDMD + IL-1 β compared to empty vector treated control (Fig. 8e, Supplementary Fig. 11c).

R4.4 Figure 2, the same suggestion, the experiment should be performed in T cell deficient mice to confirm the involvement of antitumor immunity. Additional controls should be considered, such as electroporation of empty plasmid and i.t. injection of solvent. The number of animals in each group should be increased to ensure sufficient statistical analysis.

A4.4: In new Fig. 8 we demonstrate that treatment that either includes NT GSDMD electroporation or NT GSDMD+IL-1 β electroporation is not functional in immunocompromised mice demonstrating that either pyroptosis or armed-pyroptosis treatment depends on functional immune system. In new Fig. 3, as suggested by the reviewer, we include the control where empty vector electroporated tumors were also treated with AP20187, demonstrating the importance of chemically induced pyroptosis for tumor therapy. Additional experiments including all appropriate groups were performed to enable sufficient statistical analysis.

R4.5 Figure 3, the effect of electroporated IL1b plasmid should be compared with i.t. injection of recombinant IL1b so as to better discuss the conclusion and the debate in line 235-236.

A4.5: For the purpose of our study we compared the same exposure to cytokine with or without the addition of pyroptotic component (NT GSDMD), to see if cytokine-only treatment, applied in the same way as in NT GSDMD+IL-1 β , has an antitumor effect. Previous studies have shown that IL-1 β can have

diverse effects on tumor growth, depending on the amount used and the strategy to introduce IL1 β or induce IL-1 β activation. We think that the low efficiency of plasmid electroporation in large tumors and single electroporation are the main reasons why a single cytokine-only treatment is not functional in our setup and we have strengthened the results and discussion on this issue.

R4.6 Many in vivo experimental comparisons were performed cross experiments, for example the data obtained in Figure 2 and 3 were compared directly with Figure 1 (line 200, line 229-230), instead of appropriate replication of the observations in Figure 1, this is unacceptable, especially considering the relative low mice numbers in each group. An example of this issue could be a direct inconsistency between figure 4c (where the NT-GSDMD + Isotype IgG obtained ~50% survival) and figure 1 (NT-GSDMD obtained ~25% survival)! This indicates the importance to have all indispensable controls in all experiments.

A4.6 The long-term survival of mice using different treatments is very comparable between experiments performed at different times and thus following the 3R principles we avoid repeating experiments that have been performed several times before. To clarify, in the previous Fig. 4c (now Suppl. Fig. 11 d - e), we show the data from the experiment in which we compared the the effect of NT GSDMD+IL-1 β (not NT GSDMD) in immune cell-depleted environment, so the isotype control in initially submitted Fig.4c aligns well with NT GSDMD+IL-1 β treatment (presented in initially submitted Fig. 3c, now 4c), demonstrating approximately 40% long-term survival. Taking the reviewer's advice into account, we have corrected the text to avoid a direct comparison of experiments, presented in different figures. We also performed an additional experiment with IL-12-armed pyroptosis, to compare it directly to NT GSDMD treatment alone (Fig. 5c) and show that the addition of IL-12 significantly boosts the effect of NT GSDMD-mediated pyroptosis.

R4.7 Figure 4, A, necessary labelling should be added to annotate different immune cells. Experiments in C& D miss all necessary controls (all groups without electroporation).

A4.7: Due to technical reasons and a limited panel of rare elements labeled antibodies we were unable to perform analyses of different immune cell populations using LA-ICP-MS. Instead, we performed flow cytometry of CD45+ cell populations after treatment of B16 melanoma tumors with NT GSDMD+IL-1 β electroporation compared to empty vector electroporation. We show that in armed-pyroptosis-treated tumors, CD3+ cells and NK1.1+ cells are increased compared to empty vector-treated cells. These results are shown in Fig. 8e and Supplementary Figure 11c.

In the depletion experiment, previously shown in Fig. 4c, d (now Supplementary Fig. 11d - f) we compared the effect of immune cells on the treatment efficiency of NT GSDMD+IL-1 β and we believe that we used proper controls and that following tumor growth without the electroporation in immune cell-depleted conditions is out of the scope of the study. However, to undoubtedly prove that a functional immune system is needed for tumor clearance, we exposed immunocompromised mice (New Fig. 8a - c) to the same tumor and treatment approach and showed the rapid growth of tumors regardless of the treatment (empty vector, NT GSDMD or NT GSDMD+IL-1 β).

R4.8 The isotype control for anti-PD1 and anti-CTLA4 is not correct. Clone RMP1-14 is Rat IgG2a, Clone 9D9 is a Mouse IgG2b, but the authors selected a mouse IgG2a isotype control, this should be addressed.

A4.8 We thank the reviewer for noticing the incorrect text in the methods section that included a description of an inappropriate control originating from copying the text. The IgG2a control was never performed in the context of this experiment. We corrected the text and in the revised manuscript (now this is Fig. 7) we also provide appropriate IgG controls.

R4.9 The authors described an in vivo bioluminescence method (Line 645), and bioluminescence images in the supplement figures, but it seems that is no such data provided in the main figures and text. Were all tumor sizes quantified by luminescence or volume (as depicted in all tumor growth curves in the main figures)?

A 4.9 As stated in the methods section, tumors were measured three times a week and quantified by tumor volume. The humane endpoint was set based on the tumor size measured and mice were humanly euthanized when the length, width, or height of the tumor reached 12 mm. In vivo bioluminescence method was used as a complimentary method to confirm the presence of the tumor and later on, when there was no palpation to be seen, to confirm the absence of the tumor cells and remission of the animals.

RESPONSE TO REVIEWERS

Reviewer #1 (Remarks to the Author):

My concerns were adequately addressed.

Reviewer #2 (Remarks to the Author):

The authors have addressed my concerns. I have no further questions and support the publication.

Reviewer #3 (Remarks to the Author):

The manuscript has been greatly improved and is now suitable for publication.

Reviewer #4 (Remarks to the Author):

The authors have addressed all my concerns with new experiments and improved discussions. I have no more suggestions.

Author's response to all reviewers: We would like to thank all reviewers for constructive comments that helped us to improve the study during the revision process.